# FLea: Improving federated learning on scarce and label-skewed data via privacy-preserving feature augmentation

## Abstract

Learning a global model by abstracting the knowledge, distributed across multiple clients, without aggregating the raw data is the primary goal of Federated Learning (FL). Typically, this works in rounds alternating between parallel local training at several clients, followed by model aggregation at a server. We found that existing FL methods under-perform when local datasets are small and present severe label skew as these lead to over-fitting and local model bias. This is a realistic setting in many real-world applications. To address the problem, we propose *FLea*, a unified framework that tackles over-fitting and local bias by encouraging clients to exchange privacy-protected features to aid local training. The features refer to activations from an intermediate layer of the model, which are obfuscated before being shared with other clients to protect sensitive information in the data. *FLea* leverages a novel way of combining local and shared features as augmentations to enhance local model learning. Our extensive experiments demonstrate that *FLea* outperforms the start-of-the-art FL methods, sharing only model parameters, by up to 17.6%, and FL methods that share data augmentations by up to 6.3%, while reducing the privacy vulnerability associated with shared data augmentations.

## 1 Introduction

Federated learning (FL) extracts knowledge from segregated data silos into a global model, avoiding the need to centralize the data in a single repository. The learning process is achieved through iterations between local model training and global model aggregation. Such inherent characteristics of decentralized training make FL highly suitable for privacy-sensitive applications like healthcare and finance (Rieke et al., 2020; Li et al., 2020a; Nguyen et al., 2021).

The most widely used aggregation strategy in FL is *average aggregation* (McMahan et al., 2017), where the local model parameters are averaged based on weights proportional to the size of the local data. Typically, lower aggregation weights are assigned to local models trained on smaller datasets, indicating their relatively weaker performance and lesser contribution to the global model. Nevertheless, one problem arising from such an approach is the inability to handle *data scarcity*, which refers to the situation where all clients have a limited number of samples. When all clients possess small-sized datasets, aggregating the models trained on such scarce data could become unreliable: In Sec. 3.2, we show that in the presence of data scarcity, local models are more susceptible to overfitting, i.e., they fit training samples very well but struggle to generalize to unseen testing data even if from the same distribution. In such cases, aggregating these models does not improve the global model's generalization ability. This, in turn, slows down convergence and negatively impacts the performance of the global model.

The problem could become even more pronounced when the local label distribution varies, a phenomenon referred to as *label skew*. This is a common scenario in real-world FL applications (Zhao et al., 2018; Zhu et al., 2021). Label skew alone can lead to model bias: local models are over-fitted by the local distribution and struggle to generalize to the global distribution. This is known as *client drift, which consequently leads to a sub-optimal global model* (Li et al., 2020b; Karimireddy et al., 2020; Luo et al., 2021). When local datasets are both small and label-skewed, local models are likely to fail to generalize to both in-local and out-of-local distributions. Aggregating these models does not help. It is prevalent for data scarcity and label skew to occur concurrently in the real world, such

as when clients are edge devices collecting data only about a small group of individuals (Nguyen et al., 2022) or within a limited geographical location (Karimireddy et al., 2021). While numerous studies aim to address the label skew problem in FL, the feasibility of those methods when dealing with skewed and scarce data simultaneously is still under-explored.

We categorize those FL methods into two groups: *loss-based* and *data augmentation-based* methods. The first set of methods aims to regularize the local models by modifying the local training loss to mitigate bias (Li et al., 2020b; Shi et al., 2022; Zhang et al., 2022a; Lee et al., 2022). However, these methods face sig-

Table 1: Comparing existing methods to ours.

|  | Label skew | Data scarcity | Privacy |
|---|---|---|---|
| Loss-based | ✓ | ✗ | ✓ |
| Data-based | ✓ | ✓ | ✗ |
| Our proposed | ✓ | ✓ | ✓ |

nificant challenges when dealing with limited local data since they require a balance between local optimization and global knowledge preservation. In Sec. 3.1, we demonstrate that the state-of-the-art loss-based methods under-perform as data scarcity increases. The second category of those methods proposes exchanging a global data proxy, such as a portion of raw data, aggregated data, or synthetic data, along with sharing model parameters (Zhao et al., 2018; Yoon et al., 2020; Liu et al., 2022). Although these methods tend to outperform the loss-based approaches in the presence of data scarcity, performance improvement often comes at the cost of added privacy vulnerabilities. For example, *FedMix* (Yoon et al., 2020) shares average (over mini-batches) data samples, which may reveal sensitive information consistently embedded in the entire batch of data (as illustrated in Sec. 3.3). As summarized in Table 1, our work, for the first time in the literature, addresses the challenges posed by data scarcity and label skew simultaneously, while preserving data privacy.

We propose **FLea**, a novel framework where clients exchange features along with the model parameters. Features here refer to activations from an intermediate layer of the model, given the input data. This main idea is that in deep neural networks, features not only are meaningful for classification but also provide an opportunity to protect the privacy associated with raw data (Vepakomma et al., 2020). In practical terms, *FLea* utilizes a global feature buffer that gathers feature-label pairs from multiple clients as a global proxy to aid local training. A novel feature augmentation approach is proposed to combine local and global features to alleviate both local over-fitting and model bias. A knowledge distillation strategy is also applied to the combined features in order to further prevent local model bias. To protect data privacy, features are shared by clients after applying some level of "obfuscation": we reduce the correlation between the features and the data, while maintaining their classification characters via a customized loss function. Although data augmentation has been explored in FL (Yoon et al., 2020; Guo et al., 2023), to the best of our knowledge, we are the first to design feature-level augmentation for FL, which enhances data privacy protection.

Our main contributions are:

- The first study on a common but under-explored scenario in FL, where all the clients possess limited and highly label-skewed data. We find that model over-fitting caused by data scarcity is under-looked by existing methods.
- A novel framework *FLea* to enable privacy-protected feature sharing in FL system. *FLea*, proposes an augmentation-based strategy to address both data scarcity and label skew.
- Extensive experiments with different levels of label skew and data scarcity show that *FLea* consistently outperforms baselines with only requiring clients to exchange a small proportion of the features. We also empirically demonstrate that the data privacy can be preserved from the shared features.

## 2 PRELIMINARIES AND RELATED WORKS

### 2.1 FEDERATED LEARNING AND FEDAVG

Formally, a FL system learns a global model from a set of collaborating clients, $K$, with each client $k$ containing a local dataset $\mathcal{D}_k$. A typical FL system works in synchronous rounds. At the start of each round $t$, the FL server broadcasts the current global model parameters $\theta^{(t-1)}$ to the randomly selected subset of the clients $\mathcal{K}^{(t)} \subseteq K$. Each client $k \in \mathcal{K}^{(t)}$ take a few optimization steps (e.g., using stochastic gradient descend) starting from $\theta^{(t-1)}$, resulting in an updated local model $\theta_k^{(t)}$. The local optimization aims to minimize the loss function $\mathcal{L}$ on local data $\mathcal{D}_k$, i.e., $\theta_k = \arg\min_\theta \mathcal{L}(\theta, \mathcal{D}_k | \theta^{(t-1)})$. Each round $t$ ends with model aggregation to derive the new

global model $\theta^{(t)}$. The most basic and popular aggregation method, *FedAvg* (McMahan et al., 2017) averages the model parameters weighted by the fraction of local data size in the clients,

$$\theta^{(t)} = \sum_{k \in \mathcal{K}^{(t)}} \frac{|\mathcal{D}_k|}{\sum_{k \in \mathcal{K}^{(t)}} |\mathcal{D}_k|} \theta_k. \tag{1}$$

## 2.2 LABEL SKEW IN FL

The above *FedAvg* has been shown to converge to the optimal model with local data $\mathcal{D}_k$ sampled from the same distribution (Li et al., 2019), also popularly known as the IID (independent and identically distributed) setting. However, when $\mathcal{D}_k$ are sampled from diverse distributions, known as non-IID setting, *FedAvg* usually produces low-performing models. Label skew is a typical non-IID FL problem for classification (Zhao et al., 2018; Zhu et al., 2021; Guo et al., 2023).

To mitigate the client drift, methods were proposed to *improve loss function* $\mathcal{L}$. For classification, $\mathcal{L}$ typically represents the cross-entropy loss between the target class and the prediction, termed as $\mathcal{L}_{clf}$. To compensate for missing categories in the local data, an additional regularization *Fed-Decorr* (Shi et al., 2022) discovered that severe data heterogeneity leads to dimensional collapse in FL models, prompting the introduction of regularization techniques to address this issue, $\mathcal{L}_r$, can be included. *FedProx* (Li et al., 2020b) regulates the discrepancy between the local and global model parameters. *FedNTD* (Lee et al., 2022) penalizes changes in the *logit* distribution predicted by global and local models. This penalty is applied to classes excluding the ground truth class for each sample, striking a balance between local learning and global knowledge preservation. *MOON* (Li et al., 2021) leverages constructive learning to maximize the distance between low-dimensional features and other classes, thereby improving feature learning. In addition to those methods proposing a new term $\mathcal{L}_r$, *FedLC* (Zhang et al., 2022a) directly re-scales the logits to derive a calibrated $\mathcal{L}_{clf}$. This calibration effectively mitigates classifier bias, leading to enhancements in the final global model.

Being orthogonal to a bias-agnostic loss function $\mathcal{L}$, *data augmentations* were also developed. *Zhao* et al. proposed to share a small proportion of local data globally, alongside the model parameters, to enhance *FedAvg* (Zhao et al., 2018). For example, globally sharing $5\%$ of the data yielded an improvement of up to $20\%$. For ease of presentation, we named this method *FedData* throughout the paper. Despite the desirable performance gains brought by *FedData*, collecting private data would compromise the privacy-preservation benefits of FL. Other global proxies that are less privacy-sensitive than raw data have been explored as alternatives. *FedMix* (Yoon et al., 2020) and *FedBR* (Guo et al., 2023) average data over mini-batches and share this aggregated data globally, while *CCVR* (Luo et al., 2021) shares low-dimensional features with the server to calibrate the global model on the server side. These low-dimensional features are also known as class prototypes, which are explored to mitigate local classifier bias (Tan et al., 2022b). *FedGen* (Liu et al., 2022) and VHL (Tang et al., 2022) generate some random but separable data samples to aid local learning. Although no raw information is shared by those two approaches, their performance highly depends on the quantity and quality of synthetic data, usually yielding marginal gains over *FedAvg*.

The above-mentioned methods are developed to prevent the global model from diverging under label skew, while another way to cope with this issue is to learn a personalized model per client, with the goal of enhancing performance within their local data distribution (Kulkarni et al., 2020; Tan et al., 2022a; Kotelevskii et al., 2022). Recently, personalized FL methods based on variational Bayesian inference have shown promising results, supported by theoretical guarantees (T Dinh et al., 2020; Zhang et al., 2022b; Zhu et al., 2023). However, these methods still face challenges in learning an optimal global model.

## 2.3 DATA SCARCITY IN FL

Data scarcity is a common yet under-explored challenge in FL. As first observed by (Li et al., 2022) (c.f. Finding 7), the accuracy of *FedAvg* and *FedProx* decreases as the number of clients increases (reducing local data size). Despite the proliferation of FL methods, including the methods mentioned earlier, most of them are evaluated on large local datsets, each possessing thousands of samples. This leaves their effectiveness in handling data scarcity unclear. Although experimental studies target on scarce setting (Charles et al., 2021; Zhu et al., 2023), they failed to justify if the performance gain is from alleviating the bias or the overfitting caused by data scarify, thus lack in providing a deep understanding. *FedScale* introduced the first benchmark featuring thousands of clients with limited training data (Lai et al., 2022). However, *FedScale* primarily focuses on system efficiency and offers

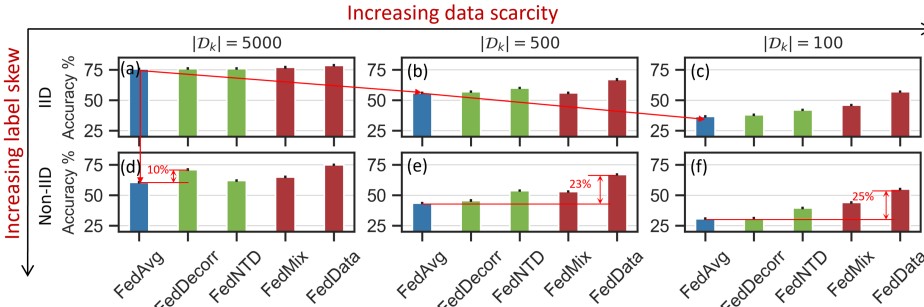

Figure 1: Performance of FL methods with varying label skew and data scarcity levels.

limited insight into algorithm effectiveness. Some client selection methods, such as discarding or reducing aggregation weights for clients with limited data (Nishio & Yonetani, 2019; AbdulRahman et al., 2020), improve overall performance while reducing communication overheads. However, these methods may not be effective when all clients face data scarcity issues.

## 2.4 PRIVACY-PRESERVING FL

Although FL aims to protect privacy via gathering model parameters instead of data, studies discovered that privacy can still be leaked from the parameters when malicious attacks exist (Dang et al., 2021; Wei et al., 2020). To enhance security, homomorphic encryption, secure aggregation, differential privacy and other techniques have been developed to defend from malicious attacks (Yin et al., 2021). These works are orthogonal to our method and can be applied in our framework. We focus on data privacy regarding the shared features. The most related works are protecting the privacy for the shared activations in splitting learning, where the server gathers activations to train part of the model (Vepakomma et al., 2020; He et al., 2020). Different from those studies, we explore FL where model training merely happens on the client side.

## 3 LIMITATION OF PREVIOUS WORK AND INSIGHTS

In this section, we reveal the limitations of previous work when applied to FL with scarce label-skewed data, and provide insights to address those limitations.

### 3.1 EMPIRICAL COMPARISON OF PREVIOUS WORK

We use CIFAR10 and compare the global model's accuracy on the global testing set. *FedAvg* is compared with the best loss-based methods *FedDecorr* and *FedNTD*, and the best data augmentation-based methods *FedMix* and *FedData*. To simulate varying label skew and scarcity levels, we split the CIFAR10 training set into 5000, 500, and 100 samples per client containing all ten classes (IID setting) or randomly chosen three classes (non-IID setting) (we refer the local data size as $|\mathcal{D}_k|$). More experimental details are presented in Appendix A.

The results are summarized in Figure 1. We draw the following conclusions: **1) *FedAvg* degrades remarkably as data scarcity and label skew increase**. Its accuracy of 75% in (a) with sufficient IID data decreases to 56% in (c) when $|\mathcal{D}_k|$ reduces to 100, and drops to 60% in (d) when the data becomes non-IID. Other methods present similar degrading trend. **2) The compared loss-based methods can address label skew only with sufficient local data**. In (d), *FedDecorr* improves *FedAvg* by 10% and performs closely to *FedData*. Yet, in (e) and (f), the advantage of *FedDecorr* disappears. While *FedNTD* consistently outperforms *FedAvg*, its accuracy still lags significantly behind *FedData*. **3) The compared data augmentation-based methods perform well at the cost of privacy leakage.** With the internal data exchange, *FedMix* and *FedData* mitigate the bias and improve local generalization simultaneously, leading to remarkable performance gains over *FedAvg*. *When the data scarcity is significant ($|\mathcal{D}_k| = 100$), they notably outperform loss-based FL methods.* However, it is worth considering that *FedMix* and *FedData* induce more privacy vulnerabilities compared to loss-based method due to sharing more data related information.

### 3.2 EFFECT OF DATA SCARCITY

To gain a deeper understanding on the impact of data scarcity, we conduct analysis in the IID setting. We empirically demonstrate that data scarcity can lead to local model over-fitting, and aggregating such models will result in the degradation of the global model.

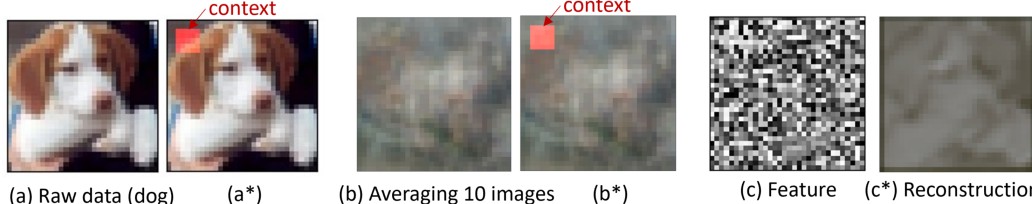

(a) Raw data (dog)    (a*)    (b) Averaging 10 images    (b*)    (c) Feature    (c*) Reconstruction

Figure 3: Data augmentations. From (a) to (c), the privacy vulnerability is reduced. (b) is the average of a batch of samples like (a), but if the local data contains individual context information (e.g., (a*)), averaging over those samples cannot protect such information (e.g., (b*)). (c) shows a feature of (a*) and (c*) shows the its reconstruction.

Following a previous study that visualizes the model bias caused by label skew using the features from the penultimate layer (Guo et al., 2023), we leverage those low-dimensional features to understand model overfitting. We look at one communication round where local training start from a global model with an accuracy of 40%. The training results are compared using different amount of local data, i.e., $|\mathcal{D}_k| = 5000|$ and $|\mathcal{D}_k| = 100|$. We employ the clustering score (Davies-Bouldin Score (DB)) to mea-

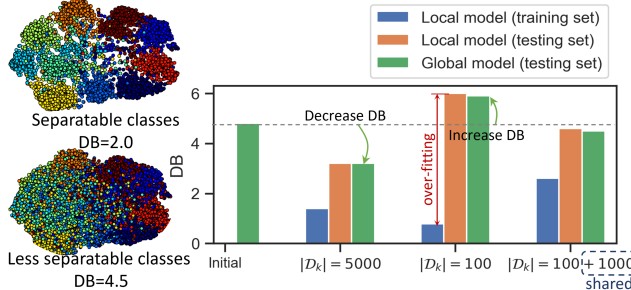

Figure 2: T-SNE for low-dimension features where the colour distinguishes classes and the class separation measurement DB under different numbers of training samples.

sure the separation of the features among classes (a detailed formulation can be found in Appendix A). As illustrated in Figure 2 (left), a smaller DB indicates less overlapped features. The DBs before and after local training are summarized in the bar chart of Figure 2. It can be observed that the gap of DB in the training set and testing set is consistently present, however the fewer the samples ($|\mathcal{D}_k| = 100$), the bigger the gap, suggesting that the over-fitting is server. Based on local models: although features for local training set are distinguishable (small DB), the features for the testing set are not (big DB). Consequently, after aggregation, the performance of the global model varies, and training with 100 samples cannot enhance the global model (an increased DB). Those results uncover that data scarcity could impact the generalization of local models, yielding overlapped features, and finally leads to under-performed global model.

As shown in the fourth group ($|\mathcal{D}_k| = 100 + 1000$), with more shared data to aid local training, the local model's generalization is improved from the third group (a smaller gap between training and testing set), and thus the global model's performance is improved (a decreased DB). This also explains why data augmentation-based methods are feasible for data scarcity in FL.

### 3.3 PRIVACY VULNERABILITY MITIGATION VIA FEATURE SHARING

Data augmentation-based methods may increase privacy vulnerability. As shown in Figure 5.3, raw data and labels are shared globally in *FedData* while aggregated data and labels are shared globally in *FedMix*. Although the average of samples in *FedMix* hinders the data reconstruction, it is still privacy vulnerable, as this will release context information. Considering an application where the client's phone has a camera sensor problem so that each photo has a spot (see Figure 5.3(a*)), or the client lives in a busy neighbourhood and thus all audio clips have a constant background score. Averaging over a batch of samples will not protect such context information, as shown in Figure 5.3(b*).

To improve the trade-offs between performance and privacy protection, we propose to share features from the intermediary layers (see Figure 5.3(c*)). We try to mitigate privacy vulnerability from three aspects: *i)* reducing the feature exposure when sharing, *ii)* hindering the data reconstruction from the features, and *iii)* increasing the difficulty of the above-mentioned context identification.

## 4 FLEA

*FLea* corrects the bias introduced by label skew and alleviates the over-fitting caused by data scarcity by using the shared features to aid local training. In *FLea*, the server will maintain a global model and a feature buffer which contains feature-target pairs from multiple clients. In the beginning, the global model is randomly initialized and the buffer is empty. Then *FLea* works in a iterative manner

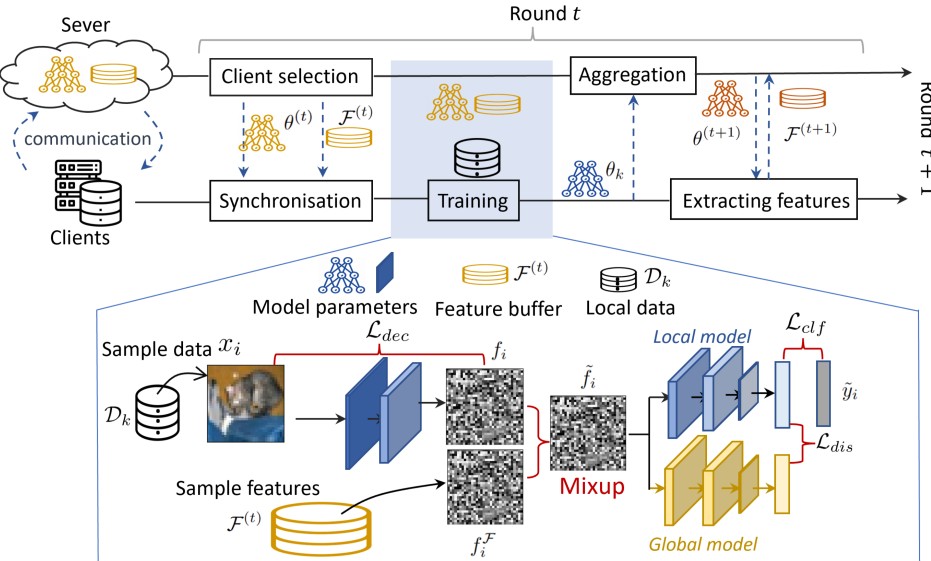

Figure 4: Overview of *FLea*, where $t$-th communication round is shown.

to updates the global model and the buffer. As illustrated in Figure 4, each round of *FLea* starts with synchronizing the global model parameters $\theta^{(t)}$ and feature buffer $\mathcal{F}^{(t)}$ to the selected clients $\mathcal{K}^{(t)}$. Once local training using $\mathcal{D}_k$ and $\mathcal{F}^{(t)}$ finishes (in the first round, only local data $\mathcal{D}_k$ is used for training since the feature buffer is empty), those clients send the updated model parameters $\theta_k$ to the server, to be aggregated into a new global model parameterized by $\theta^{(t+1)}$. *FLea* uses the same aggregation strategy as *FedAvg* (Eq. (1)). Followed by that, *FLea* needs another step to update the global feature buffer to $\mathcal{F}^{(t+1)}$. A detailed training procedure can be found in Appendix B and Algorithm 1. We elaborate on the main components in the procedure as follows.

**Feature buffer**: Let's consider the global model parameters $\theta^{(t)}$ to be divided into two parts at layer $l$: $\theta^{(t)}[: l]$ and $\theta^{(t)}[l :]$. For client $k$, the feature vector extracted from a data point $x_i \in \mathcal{D}_k$ is $\theta^{(t)}[: l](x_i) = f_i^{\mathcal{F}}$. The feature buffer from this client is the set of pairs including target labels and feature vectors $(f_i^{\mathcal{F}}, y_i^{\mathcal{F}})$. Each client randomly selects $\alpha$ fraction of its local data to create its feature buffer to share with others. The server gathers those local feature buffers and merge them into the global one $\mathcal{F}^{(t)}$. Note that a client only extracts and contributes to the global feature buffer at the round when it participates in training and the global buffer resets at every round.

**Client $k$'s local training**: Suppose client $k$ is selected in round $t$, i.e., $k \in \mathcal{K}^{(t)}$. As shown in Figure 4, $k$ receives the global model $\theta_k = \theta^{(t)}$ and the feature buffer $\mathcal{F}^{(t)}$. The local data $\mathcal{D}_k$ and the feature buffer $\mathcal{F}^{(t)}$ are divided into equal-sized batches for model optimization, termed by $\mathcal{B} = \{(x_i, y_i) \in \mathcal{D}_k\}$ and $\mathcal{B}^f = \{(f_i^{\mathcal{F}}, y_i^{\mathcal{F}}) \in \mathcal{F}^{(t)}\}$, respectively ($|\mathcal{B}| = |\mathcal{B}^f|$). The traditional method will feed $\mathcal{B}$ into the model directly to optimize the model but we propose to augment the input in the feature space. We feed $\mathcal{B}$ into the model, extracting the intermediate output for each data point: $f_i, \forall x_i \in \mathcal{B}$, and generate the augmentation as,

$$
\begin{aligned}
\tilde{f}_i &= \lambda_i f_i + (1 - \lambda_i) f_i^{\mathcal{F}}, \\
\tilde{y}_i &= \lambda_i y_i + (1 - \lambda_i) y_i^{\mathcal{F}},
\end{aligned}
\tag{2}
$$

where $(f_i, y_i)$ and $(f_i^{\mathcal{F}}, y_i^{\mathcal{F}})$ are two feature-target pairs from $\mathcal{B}$ and $\mathcal{B}^f$, respectively. Inspired by the data augmentation method in the centralized setting (Zhang et al., 2018), we sample the weight $\lambda_i$ for each data point from a symmetrical Beta distribution (Gupta & Nadarajah, 2004): $\lambda_i \sim Beta(a, a)$. $\lambda_i \in [0, 1]$ controls the strength of interpolation between the local and global feature pairs: A smaller $\lambda_i$ basically makes the generated sample closer to the local feature while a larger one pushes that to the global feature.

Following the augmentation, the training loss for each batch is designed to contain two parts: one for classification ($\mathcal{L}_{clf}$) and one for knowledge distillation form the global model ($\mathcal{L}_{dis}$). The classification loss $\mathcal{L}_{clf}$ is formulated as,

$$
\mathcal{L}_{clf}(\mathcal{B}, \mathcal{B}^f) = \frac{1}{|\mathcal{B}|} \sum_i \sum_c -\tilde{y}_i[c] \log p_i^l[c],
\tag{3}
$$

where for $\tilde{f}_i$, the logit is $z_i^l = \Gamma_{\theta_{k,l:}}(\tilde{f}_i)$ and the probability for class $c$ is $p_i^l[c] = \frac{exp(z_i^l[c])}{\sum_c exp(z_i^l[c])}$. The distillation loss (Hinton et al., 2015) is derived by the KL-divergence between the global probabilities and local probabilities as,

$$\mathcal{L}_{dis}(\mathcal{B}, \mathcal{B}^f) = \frac{1}{|\mathcal{B}|} \sum_i \sum_c -p_i^l[c] \log \frac{p_i^g[c]}{p_i^l[c]}, \quad (4)$$

where for $\tilde{f}_i$ the global logit is $z_i^g = \Gamma_{\theta_{l:}^{(t)}}(\tilde{f}_i)$ and the global probability is $p_i^g[c] = \frac{exp(z_i^g[c])}{\sum_c exp(z_i^g[c])}$. Meanwhile, we aim to obfuscate the features to protect data privacy before they are shared out from the clients. As such, we learn the $l$ layers while reducing the correlation between the features and the source data. This is achieved by minimizing the loss (Vepakomma et al., 2020) below,

$$\mathcal{L}_{dec}(\mathcal{B}) = \frac{Tr(X^T F F^T X)}{\sqrt{Tr(X^T X)^2}\sqrt{Tr(F^T F)^2}}, \quad (5)$$

where $X \in \mathbb{R}^{|\mathcal{B}| \times d}$ and $F \in \mathbb{R}^{|\mathcal{B}| \times d^f}$ are the data and feature matrix. Note that each $X_i \in \mathbb{R}^d$ and $F_i \in \mathbb{R}^{d^f}$ are the flattening vector for data $x_i$ and feature $f_j^l$. The numerator of $\mathcal{L}_{dec}$ measures the covariance between the data and the features, while its denominator measures the averaged pairwise distance within the data batch and feature batch, respectively. When updating the local model, the features change correspondingly. It is desired that the distance covariance decreases faster than the feature inner distance for each batch. Since when reducing the correlation, we hope the features can maintain classification ability, and thus we optimize all the loss functions jointly, as follows,

$$\mathcal{L} = \mathcal{L}_{clf}(\mathcal{B}, \mathcal{B}^f) + \lambda_1 \mathcal{L}_{dis}(\mathcal{B}, \mathcal{B}^f) + \lambda_2 \mathcal{L}_{dec}(\mathcal{B}), \quad (6)$$

where $\lambda_1$ and $\lambda_2$ are the weights to trade-off classification and privacy preserving. The local update is then achieved by $\theta_k \leftarrow \theta_k - \eta \frac{\partial \mathcal{L}}{\partial \theta_k}$, where $\eta$ controls the learning rate.

**Feature buffer updating**: After the global model aggregation and broadcasting, client $k$ extracts the features from the new model parameterized by $\theta^{(t+1)}$ from layer $l$ to formulate the feature set. Those sets will be sent to the server to replace the old ones, updating the feature buffer to $\mathcal{F}(t+1)$. The iterations continue until the global model converges.

## 5 EVALUATION

### 5.1 EXPERIMENTAL SETUP

**Datsets.** We evaluate FLea on two different data modalities, classifying images (CIFAR10 Krizhevsky et al. (2009)) and audio (UrbanSound8K Salamon et al. (2014)). Both datasets have 10 classes. Following strategies from (Zhang et al., 2022a), we distribute the training data to $|K|$ clients via quantity-based skew (Quantity($q$)) and distribution-based skew (Dirichlet($\mu$)) splits. More details can be found in the Appendix C.1.

**Model architectures and hyper-parameters.** We classify images in CIFAR10 using MobileNet_V2 (Sandler et al., 2018) that has 18 blocks consisting of multiple convolutional and pooling layers. *FLea* shares the features after the first block as illustrated in Table 4. For audio classification, the samples are first transformed into spectrograms and fed into a 4 layer CNN model, which we termed as *AudioNet*. We share the feature from the second convolutional layer and the details of architecture can be found in Table 5.

We use the Adam optimizer for local training with an initial learning rate of $10^{-3}$ and decay it by $2\%$ per communication round until $10^{-5}$. The size of the local batch is 64, and we run 10 local epochs for 100 clients setting for CIFAR10 and 15 local epochs for other settings. $10\%$ of clients are randomly sampled at each round. We run 100 communications and take the best accuracy as the final result. Without particular mention, we use $\lambda \sim Beta(2, 2)$ for Eq. (2), and $\lambda_1 = 1, \lambda_2 = 3$ for the loss in Eq. (6). For all results, we report the mean and standard deviation of the accuracy from five runs with different random seeds. More details are presented in Appendix C.2.

**Baselines.** We compare *FLea* against *FedAvg*, and then *loss-based* methods: $i$) *FedProx* (Li et al., 2020b), $ii$) *FedDecorr* (Shi et al., 2022), $iii$) *FedLC* (Zhang et al., 2022a), and $iv$) *FedNTD* (Lee et al., 2022), as well as *data augmentation-based* methods: $i$) *FedBR* (Guo et al., 2023), $ii$) *CCVR* (Luo et al., 2021), $iii$) *FedGen* (Liu et al., 2022), $iv$) *FedData* (Zhao et al., 2018) and $v$) *FedMix* (Yoon et al., 2020). All baselines are hyper-parameter optimized to report their best performances. The specific setting can be found in Appendix C.3.

Table 2: Overall performance comparison. The local data size $|\mathcal{D}_k|$ is as small as 100 on average. Accuracy is reported as $mean \pm std$ across five runs. The best baseline (excluding *FedData*) under each column is highlighted by red and the second best is highlighted by grey.

| Accuracy % | CIFAR10 ($|K| = 500$) | | | UrbanSound8K ($|K| = 70$) | | |
|---|---|---|---|---|---|---|
| | $Quantity(3)$ | $Dirichlet(0.5)$ | $Dirichlet(0.1)$ | $Quantity(3)$ | $Dirichlet(0.5)$ | $Dirichlet(0.1)$ |
| FedAvg | 30.25±1.33 | 32.58±1.09 | 20.46±2.15 | 43.69±0.56 | 46.77±0.87 | 34.59±2.64 |
| FedProx | 31.92±1.45 | 32.01±1.25 | 20.86±1.97 | 38.45±0.48 | 39.58±1.02 | 34.81±0.46 |
| FedDecorr | 31.12±1.57 | 33.57±1.22 | 21.34±1.59 | 45.01±0.57 | 46.77±0.65 | 35.87±1.03 |
| FedLC | 32.05±1.60 | 30.17±1.18 | 18.82±2.01 | 50.98±0.49 | 50.11±0.83 | 37.05±0.87 |
| FedNTD | 39.98±0.97 | 39.82±0.86 | 26.78±2.34 | 49.80±0.45 | 51.09±0.97 | 36.53±0.99 |
| FedBR | 31.66±1.07 | 33.08±1.12 | 20.98±2.54 | 44.05±0.63 | 47.58±0.90 | 36.15±1.17 |
| CCVR | 35.95±1.63 | 35.02±1.43 | 24.21±2.67 | 47.12±0.72 | 49.26±0.92 | 39.62±1.20 |
| FedGen | 32.32±1.21 | 34.27±1.56 | 22.56±2.89 | 45.20±0.89 | 48.33±1.12 | 38.27±1.44 |
| **Each client shares 10% of the data or features** | | | | | | |
| FedData | 54.64±1.02 | 56.47±1.22 | 55.35±1.46 | 62.83±1.25 | 64.45±0.76 | 61.11±0.98 |
| FedMix | 44.04±1.53 | 45.50±1.88 | 38.13±2.06 | 51.56±0.59 | 54.18±0.62 | 43.35±0.72 |
| **FLea** | 47.03±1.01 | 48.86±1.43 | 44.40±1.23 | 57.73±0.51 | 59.22±0.78 | 45.94±0.77 |

## 5.2 RESULTS

**Overall accuracy.** We summarize the overall comparison to baselines in Table 2, and present extended results in Appendix C.4. *FLea* consistently outperforms the baselines and closes the gap to *FedData* across different data scarcity and label skew levels for the two tasks.

Among the compared baselines that only share model parameters, *FedNTD* achieves the best performance in most cases, but it is outperformed by *FLea* by $7.03 \sim 17.62\%$ across all the settings. Notably, the most significant improvement occurs when the label skew intensifies, specifically with the $Dirichlet(0.1)$ distribution. This suggests that *FLea*, by utilizing features from other clients to compensate for locally absent distributions, is more effective in mitigating local bias. Furthermore, we observe that *FLea* exhibits faster learning compared to the non-augmentation methods. Illustrated in Figure 5, *FLea* consistently requires fewer communication rounds to attain a target model accuracy. Although this does not necessarily imply superior communication efficiency for *FLea*, as additional features need to be transferred and stored. *FLea* still proves advantageous in scenarios where extensive communication with a large number of clients is not always feasible.

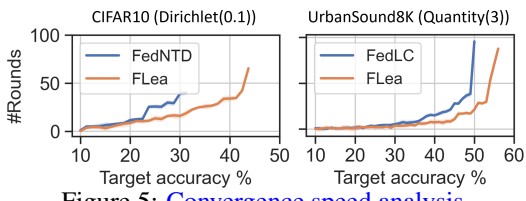

Figure 5: Convergence speed analysis.

Figure 6: Results for sharing different fractions of features/data.

Compared to the augmentation-based counterparts, *FLea* is also superior. *FedMix* is the state-of-the-art data augmentation-based approach, yet *FLea* surpasses *FedMix* with a performance gain ranging from 2.59% to 6.27%. Besides, we present the results for sharing different fractions of the features in Figure 6. It is evident that a fraction of 10% can significantly boost the performance while when sharing more, the advantage over *FedMix* still keeps. All those demonstrate the superiority of employing feature augmentation across all classes to enhance local model generalization. In Sec. 5.3, we will further show that *FLea* is also more privacy-perserving than *FedMix*.

**Impact of hyper-parameters.** We illustrate how we identify the hyper-parameters $\lambda_1$ and $\lambda_2$ for the loss function and $a$ in the $Beta$ distribution for the augmentation in Figure 7. We first set $\lambda_2 = 0$ (without obfuscating the features) and search the value for $\lambda_2$. As shown in Figure 7(a), we found that $\lambda_1 > 1$ can improve the performance compared to that without the distilling loss ($\lambda_1 = 0$), but if the weight is too large ($\lambda_1 > 4$) it harms the performance. The pattern is similar with other $\lambda_2$, and thus we informally use $\lambda_1 = 1$ for all experiments. With $\lambda_1 = 1$, we further study how $\lambda_2$ impacts the trade-off between privacy preservation (reflected by the reduced correlation) and the feature utility (reflected by the model accuracy), as shown in Figure 7(b). Enlarging $\lambda_2$ can significantly enhance privacy protection (referring to the increasing $1 - \bar{c}$) but decreases the final performance. We finally use $\lambda_2 = 3$ when the $\bar{c}$ reduces to about 0.72 while maintaining a strong accuracy of about 57%. We also suggest future applications using $2 \sim 6$ for the trade-off. In Figure 7(c), we demonstrate that the final performance is not sensitive to the parameter of the $Beta$ distribution since we always have an expectation of 0.5 for $\lambda$.

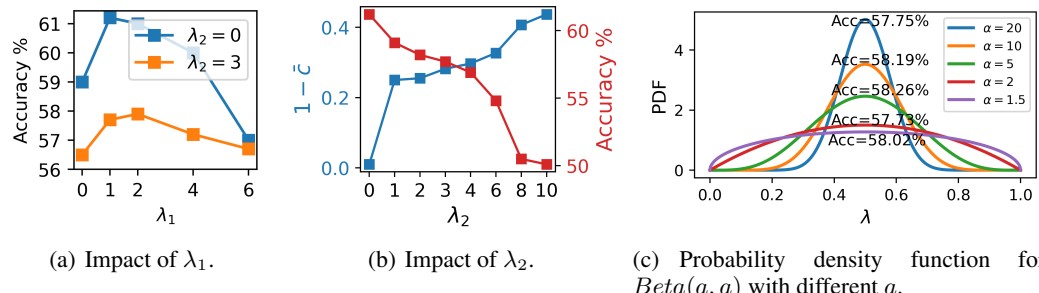

(a) Impact of $\lambda_1$.  (b) Impact of $\lambda_2$.  (c) Probability density function for $Beta(a, a)$ with different $a$.

Figure 7: Hyper-parameter tuning for *UrbanSound8K, $|K| = 70$ under $Quantity(3)$ split*. In (b), $\bar{c}$ presents the averaged correlation between the feature and the original data for the 100 rounds.

### 5.3 FLEA MITIGATES PRIVACY RISK

*FLea* aims to mitigate the privacy leakage from three aspects: *i)* reducing feature exposure by only sharing features with a fraction of clients, *ii)* defending from data reconstruction attack, and *iii)* preventing the sensitive context information from being identified. We demonstrate *FLea* is more privacy-preserving than *FedMix* and *FedData* as follows.

***Feature exposure.*** To quantify the feature exposure, we define a feature exchange matrix $\xi \in \mathbb{R}^{|K| \times |K|}$. $\xi_{i,j}^{(t)} = 1$ denotes client $i$ and $j$ have exchanged features (from $\alpha$ fraction of local data) for at least once until (including) $t$-th round, otherwise $\xi_{i,j}^{(t)} = 0$. The the feature exposure is measured by

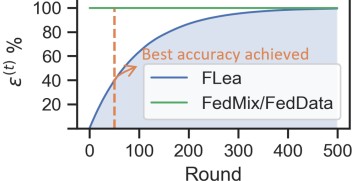

$\epsilon^{(t)} = \sum_{i,j} \xi_{i,j}^{(t)} / K^2$ ( $0 \leq \epsilon^{(t)} \leq 1$), and a smaller $\epsilon^{(t)}$ is better. As *FedData* and *FedMix* gather data or data averages and broadcast the data to all clients before local model training, $\epsilon^{(t)} = 100\%$ consistently. We illustrate

Figure 8: Feature exposure with communication rounds.

$\epsilon^{(t)}$ for *FLea* in Figure 8. From the figure it can be observed that the exposure of *FLea* grows slowly. In our experiments, the model converges within 50 rounds, by when $\epsilon^{(t)} \leq 40\%$.

It is worth noting that feature exposure is not equivalent to privacy leakage, as the features of *FLea* do not leak source data. As shown in Figure 7(b), our loss function with $\mathcal{L}_{dec}$ can effectively reduce the correlation between features and raw data, so that the privacy is not compromised. To demonstrate that *FLea* can defend from data reconstruction and context identification attacks, we construct testbeds by assuming the attacker can access the entire *CIFAR10* training set. The setup and results are summarized in Appendix C Sec. D, and the main findings are given blew.

***Data reconstruction***. Assume one client is selected to share the feature of an image, e.g., the dog image in Figure (a), in a certain communication round (when $c = 0.4$), an optimal attacker tries to reconstruct the image from the feature and it will end up to the image as shown in Figure (c*). The original attribute, i.e., the color distribution cannot be recovered and thus privacy is preserved.

***Identifying context information***. As we mentioned before, one advantage of *FLea* over *FedMix* is that *FLea* can better protect the context information. Again using the example, if one client is selected to share some augmentations from the data including the dog image in Figure (a*), the context attacker can easily detect the context marker in Figure (b*) shared by *FedMix*, but can be difficult to detect that from the feature in Figure (c) shared by *FLea*. Therefore, the context privacy is better preserved.

## 6 CONCLUSIONS

We proposed *FLea*, a novel approach to tackle scarce and label-skewed data in FL. Feature augmentation is employed to mitigate over-fitting and local bias simultaneously. Extensive experiments demonstrate that *FLea* remarkably outperforms the state-of-the-art baselines.

*Limitations.* In reality, because of extra feature sharing, *FLea* can introduce some extra overheads like communication and storage. We leave the improvement of the efficiency for real-world applications for future work. To enhance privacy, *FLea* can be trivially combined with methods protecting model parameters (such as DP-SGD (Abadi et al., 2016)), however, improving the statistical privacy risks posed by feature sharing is hard due to the challenges originating from the stochasticity in modeling, data distribution in the clients, the high dimensionality of real-valued features, etc. We thus leave this as a future work. We anticipate that our work will inspire further investigations to comprehensively study feature sharing in the low data regime.

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

APPENDIX

## A   EXPERIMENTAL SETUP FOR SECTION 3

**Federated Learning Setup.** In *Sec. 3 Limitations of Previous Work and Insights*, we employ CIFAR10 for an empirical comparison. We conduct six groups of experiments to simulate different levels of label skew and scarcity, as introduced below.

- **(a)** The training set of CIFAR10 is uniformly distributed to 10 clients, resulting in each local dataset having a size of 5000 ($|\mathcal{D}_k| = 5000$) and 10 classes (IID). More specifically, each local dataset has 500 samples per class.

- **(b)** CIFAR10 is uniformly distributed to 100 clients. Thus each local dataset has a size of 500 ($|\mathcal{D}_k| = 500$), 10 classes (IID) with each class containing 50 samples.

- **(c)** CIFAR10 is uniformly distributed to 500 clients. Each local dataset have a size of 100 ($|\mathcal{D}_k| = 100$), 10 classes (IID) with each class containing 10 samples.

- **(d)** The training set of CIFAR10 is distributed to 10 clients, but only three out of ten classes are distributed to each client (non-IID), resulting in each local dataset having a size of 5000 ($|\mathcal{D}_k| = 5000$) and approximately 1666 samples for each class.

- **(e)** Similar to (d), CIFAR10 is distributed to 100 clients ($|\mathcal{D}_k| = 500$) with each client containing 3 classes (non-IID). This data distribution is shown in Figure 9(a).

- **(f)** Similar to (d), CIFAR10 is distributed to 500 clients ($|\mathcal{D}_k| = 100$). Each client contains 3 classes (non-IID) with about 33 samples per class.

For CIFAR10 classification, we employ MobileNet_V2, which has 18 blocks consisting of multiple convolutional and pooling layers (Sandler et al., 2018). We use the Adam optimizer for local training with an initial learning rate of $10^{-3}$ and decay it by 2% per communication round until $10^{-5}$. For (a) and (d), all clients will participate in the training in each round, while for the other groups, we will randomly select 10% of the clients for each round. The size of the local batch is 64, and we run 10 local epochs for groups (a, b, d, e) and 5 local epochs for groups (c, f). We run 100 communication rounds for all groups to ensure global convergence.

**Experimental setup for Figure 1:** To compare the performance of existing methods with , we use CIFAR10 dataset and report the classification accuracy of the global model based on the global testing set. We compare *FedAvg* with loss-based methods such as *FedDecorr* and *FedNTD*, as well as data augmentation-based methods like *FedMix* and *FedData*. They are the most representative methods in each category. *FedMix* is implemented by averaging every 10 samples and sharing the result globally. The shared averaged data is then combined with local data according to a Beta distribution (with the $a = 2$) for local training. In the case of *FedData*, we collect 10% of the data (randomly chosen) from each client and share it globally, in the first communication round. To simulate varying scarcity levels, we split the CIFAR10 training set (comprising 50,000 samples in total) into 5000, 500, and 100 training samples on average per client, which ends up with 10, 100 and 500 clients finally. Other settings are the same with the main experiments as introduced in Sec. 5.1.

**Experimental setup for Figure 2:** DB score (Davies & Bouldin, 1979) is defined as the average similarity measuring each cluster with its most similar cluster, where similarity is the ratio of within-cluster distances to between-cluster distances. Thus, clusters which are farther apart and less dispersed will result in a better score. The minimum score is zero, with lower values indicating better clustering. To calculate the score for features, we use the ground-true class labels as cluster labels, and use Euclidean distance between features to measure the similarity.

For a fair comparison, the local training for all clients starts from a same global status with an accuracy of 40%. The features of the testing set from the initial global model present a DB of 4.8. We run one communication round and report the performance for the global model. In this round, for $|\mathcal{D}_k| = 5000$ we aggregate 10 clients while for $|\mathcal{D}_k| = 100$ we aggregate 50 clients, so that the total samples used for model training are kept unchanged. For $|\mathcal{D}_k| = 100 + 1000$ group, we additionally give the selected 50 clients 1000 samples (gathered in the first round) to aid local training. In Figure 2, for local models, we report the averaged DB across clients.

## B NOTATIONS AND ALGORITHM

Table 3: Notations used in this paper.

| | | | |
|---|---|---|---|
| $\|\mathcal{C}\|$ | The number of classes | $T$ | The number of rounds |
| $\|K\|$ | The number of clients | $k$ | Client $k$ |
| $\mathcal{D}_k$ | client $k$'s data | $\mathcal{D}$ | Global data |
| $l$ | From layer $l$ to extract features | $\theta^{(t)}$ | Global model at $t$-th round |
| $\theta_{k,:l}$ | Top $l$ layers of the model | $\theta_{k,l:}$ | Layers after $l$ of the model |
| $\mathcal{F}_k^{(t)}$ | Local feature set | $\mathcal{F}^{(t)}$ | Global feature set |
| $a$ | Parameter for Beta distribution | $\lambda$ | Weigh for feature mix-up in loss |
| $\alpha$ | Local feature sampling fraction | $\lambda2, \lambda3$ | Weigh in loss |

---

**Algorithm 1:** Federated Learning with Feature Sharing (FLea)

---

**Input** : Total rounds $T$, local learning rate $\eta$, local training epochs $E$, sampled clients set $\mathcal{K}^{(t)}$,
        a given layer $l$, parameter $a$ for Beta distribution.
**Output:** Global model $\theta^{(T)}$.

1 **Initialize $\theta^{(0)}$ for the global model**
2 **for** *each round $t$ = 1,2,...,T* **do**
3      Server samples clients $\mathcal{K}^{(t)}$ and broadcasts $\theta_k \leftarrow \theta^{(t-1)}$
4      Server broadcasts the feature $\{\mathcal{F}^{(t)}, .., \mathcal{F}^{(t-\tau)}\}$ to clients in $\mathcal{K}^{(t)}$    // Skip if $t = 1$.
5      **for** *each client $k \in \mathcal{K}^{(t)}$ in parallel* **do**
6          **for** *local step $e = 1, 2, .., E$* **do**
7              **for** *local batch $b = 1, 2, ...$* **do**
8                  sample $\lambda \sim Beta(a, a)$
9                  $\theta_k \leftarrow \theta_k - \eta\nabla\mathcal{L}(\theta_k)$   // if $t = 1$, only use local data for training. Otherwise,
                        use one batch of local data $\mathcal{D}_k$ and one batch of global feature $\mathcal{F}^{(t)}$ according
                        to Eq. (6).
10              **end**
11          **end**
12          Client $k$ sends $\theta_k$ to server
13      **end**
14      Server aggregates $\theta_k$ to a new global model $\theta^{(t)}$ refer to Eq. (1)
15      **for** *each client $k \in \mathcal{K}^{(t)}$ in parallel* **do**
16          Client $k$ receives model $\theta^{(t)}$
17          Client $k$ extracts (without gradients) and sends $\mathcal{F}_k^{(t)}$ to server
18      **end**
19 **end**

---

**Beta Distribution.** The probability density function (PDF) of the Beta distribution is given by,

$$f(\lambda; a, b) = \frac{\lambda^{a-1}(1-\lambda)^{(b-1)}}{N}, \tag{7}$$

where $N$ is the normalizing factor and $\lambda \in [0, 1]$. In our study, we choose $a = b$ and herein,
$f(\lambda) = \frac{1}{N}(\lambda(1-\lambda))^{a-1}$.

## C DETAILS OF EXPERIMENTS

### C.1 DATA DISTRIBUTION

**Image data:** We test our algorithm on CIFAR10 (Krizhevsky et al., 2009). We distribute CIFAR10 training images (containing $50,000$ samples for 10 classes) to $K = 100$ and $K = 500$ clients and use the global CIFAR10 test set (containing $1,000$ samples per class) to report the accuracy of the global model. We show the data splits for 100 clients setting in Figure 9. For 500 clients setting

the distribution is similar, but the number of samples per client reduces to one-tenth of the number shown in Figure 9.

**Audio data:** We also test *FLea* using UrbanSound8K dataset Salamon et al. (2014). This dataset contains 8732 labeled sound excerpts ($\leq 4s$) of urban sounds from 10 classes: air conditioner, car horn, children playing, dog bark, drilling, engine idling, gun shot, jackhammer, siren, and street music. For experiments, we randomly hold out 20% (about 1700 samples) for testing and distribute the rest (about 7000 samples) to $K$ clients for training. We report the results for $K = 70$ and $K = 140$, using the $Quantity(3)$, $Dirichlet(0.5)$, and $Dirichlet(0.1)$ splits.

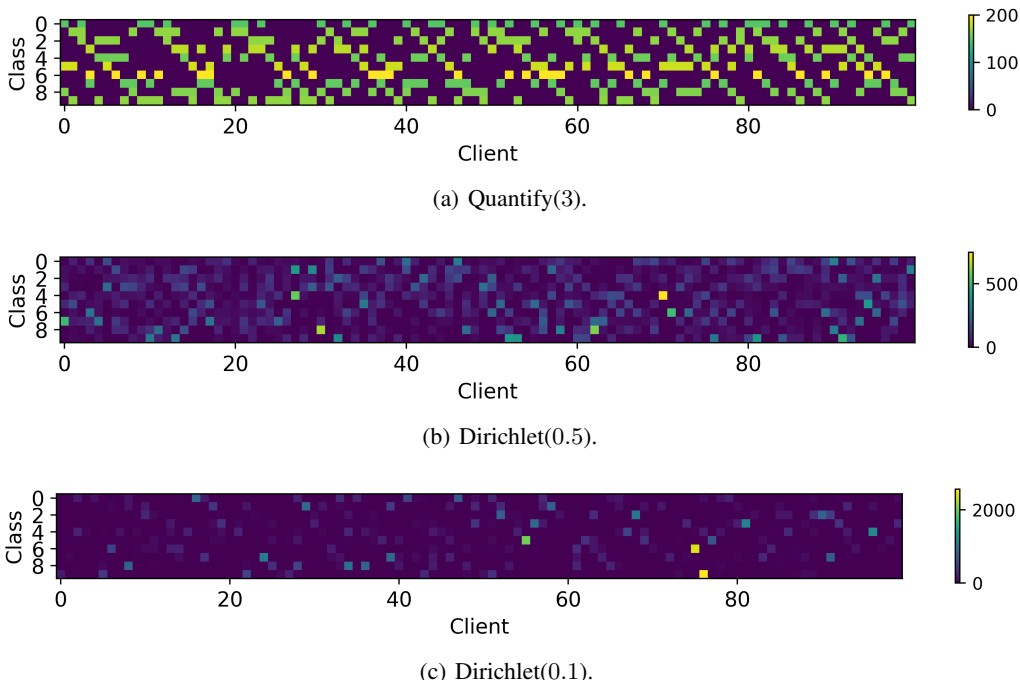

(a) Quantify(3).

(b) Dirichlet(0.5).

(c) Dirichlet(0.1).

Figure 9: Training data split for CIFAR10, $|K| = 100$.

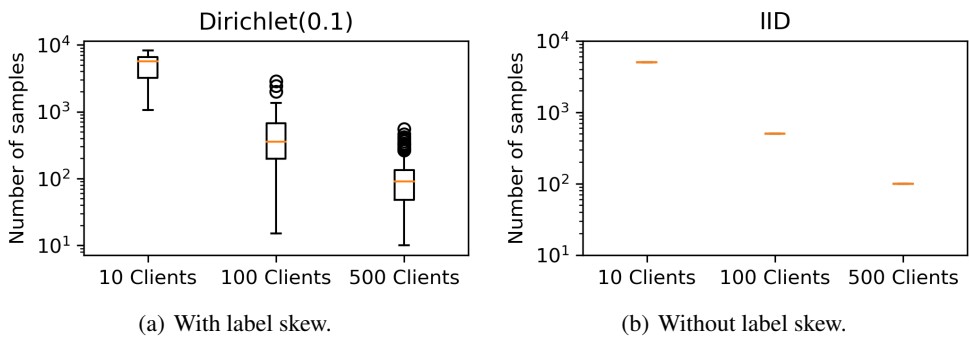

(a) With label skew.       (b) Without label skew.

Figure 10: The distribution of the number of samples per client $|\mathcal{D}_k|$ for CIFAR10.

To better illustrate the data scarcity problem, we visualize the distribution for the local data size in Figure 10. As it shown, when we distribute the training set of CIAFR10 (50, 000 samples) to 10 clients using a Dirichlet distribution parameterized by 0.1, these clients will present different class distributions, and the total number of local samples ranges from 2853 to 8199. This is the commonly explored non-IID setting. In this paper, we further explore scarce non-IID data, and thus we split the data into 100 and 500 clients. As a result, the number of samples per client reduces significantly: the median number drops from 5685 to 90 when the number of clients increases from 10 to 500,

as shown in Figure 10(a). This is the realistic scenario that we are interested in. It is also worth mentioning that data scarcity is independent of label skew and it can happen in the IID scenario. As shown in Figure 10(b), the local data covers 10 classes uniformly, but the data scarcity problem becomes severe when the number of client increase.

## C.2 MODEL ARCHITECTURE AND HYPER-PARAMETERS

We classify images in CIFAR10 using MobileNet_V2 (Sandler et al., 2018) that has 18 blocks consisting of multiple convolutional and pooling layers. The architectures of MobileNet_V2 for CIFAR10 is summarized in Table 4.

For audio classification, the audio samples are first transformed into spectrograms and fed into a CNN model, which we termed as *AudioNet*. This model consists of four blocks, each comprising a convolutional layer, a Relu activation layer, and a Batch Normalization layer, followed by a fully connected layer[1]. The details of the convolutional layers are summarized in Table 5.

We use the Adam optimizer for local training with an initial learning rate of $10^{-3}$ and decay it by $2\%$ per communication round until $10^{-5}$. The size of the local batch is $64$, and we run 10 local epochs for 100 clients setting and 15 local epochs for the rest. For feature augmentation, we use $Beta(2, 2)$. The weights in the loss function are set to $\lambda_1 = 1$ and $\lambda_2 = 3$. $10\%$ of clients are randomly sampled at each round. We run 100 communications and take the best accuracy as the final result. For all results, we report the mean and standard deviation of the accuracy from five runs with different random seeds.

Table 4: Architecture of MobileNet_V2. Features used to report the results in Table 6 are underlined.

| Block(CNN layers) | #Input | Operator | #Output Channel | #Kernel | #Stride | #Output |
|---|---|---|---|---|---|---|
| $0(1)$ | $3 \times 32 \times 32$(image) | conv2d | $32$ | $3$ | $1$ | $32 \times 32 \times 32$ |
| $1(2-5)$ | $32 \times 32 \times 32$ | conv2d$\times 4$ | $32, 32, 16, 16$ | $1, 3, 1, 1,$ | $1, 1, 1, 1$ | $\underline{16 \times 32 \times 32}$ |
| $2(6-9)$ | $16 \times 32 \times 32$ | conv2d$\times 4$ | $96, 96, 24, 24$ | $1, 3, 1, 1$ | $1, 1, 1, 1$ | $32 \times 32 \times 32$ |
| $3(10-12)$ | $32 \times 32 \times 32$ | conv2d$\times 3$ | $144, 144, 24$ | $1, 3, 1$ | $1, 1, 1$ | $24 \times 32 \times 32$ |
| $4(13-14)$ | $24 \times 32 \times 32$ | conv2d$\times 3$ | $144, 144, 32$ | $1, 3, 1$ | $1, 2, 1$ | $32 \times 16 \times 16$ |
| $5\&6(15-20)$ | $32 \times 16 \times 16$ | conv2d$\times 3$ | $192, 192, 32$ | $1, 3, 1$ | $1, 1, 1$ | $32 \times 16 \times 16$ |
| $7(21-23)$ | $32 \times 16 \times 16$ | conv2d$\times 3$ | $192, 192, 64$ | $1, 3, 1$ | $1, 2, 1$ | $64 \times 8 \times 8$ |
| $8, 9, \&10(24-32)$ | $64 \times 8 \times 8$ | conv2d$\times 3$ | $384, 384, 64$ | $1, 3, 1$ | $1, 1, 1$ | $64 \times 8 \times 8$ |
| $11(33-36)$ | $64 \times 8 \times 8$ | conv2d$\times 4$ | $384, 384, 96, 96$ | $1, 3, , 11$ | $1, 1, 1, 1$ | $9 \times 8 \times 8$ |
| $12\&13(37-42)$ | $96 \times 8 \times 8$ | conv2d$\times 3$ | $576, 576, 96$ | $1, 3, 1$ | $1, 1, 1$ | $96 \times 8 \times 8$ |
| $14(43-45)$ | $96 \times 8 \times 8$ | conv2d$\times 3$ | $576, 576, 160$ | $1, 3, 1$ | $1, 2, 1$ | $160 \times 4 \times 4$ |
| $15\&16(46-51)$ | $160 \times 4 \times 4$ | conv2d$\times 3$ | $960, 960, 160$ | $1, 3, 1$ | $1, 1, 1$ | $160 \times 4 \times 4$ |
| $17(52-54)$ | $160 \times 4 \times 4$ | conv2d$\times 3$ | $960, 960, 320$ | $1, 3, 1$ | $1, 1, 1$ | $320 \times 4 \times 4$ |
| $18(55)$ | $320 \times 4 \times 4$ | conv2d | $1280$ | $1$ | $1$ | $1280 \times 4 \times 4$ |

Table 5: Architecture of AduioNet. Features used to report the results in Table 7 are underlined.

| Index | #Input | Operator | #Output Channel | #Kernel | #Stride | #Output |
|---|---|---|---|---|---|---|
| $1$ | $2 \times 64 \times 344$(2-channel spectrogram) | conv2d | $8$ | $5$ | $2$ | $8 \times 32 \times 172$ |
| $2$ | $8 \times 32 \times 172$ | conv2d | $16$ | $3$ | $2$ | $\underline{16 \times 16 \times 86}$ |
| $3$ | $16 \times 16 \times 86$ | conv2d | $32$ | $3$ | $2$ | $\underline{32 \times 8 \times 43}$ |
| $4$ | $32 \times 8 \times 43$ | conv2d | $64$ | $3$ | $2$ | $64 \times 4 \times 22$ |

## C.3 BASELINE IMPLEMENTATION

More details for baseline implementations are summarized as blew,

- *FedProx*: We adapt the implementation from (Li et al., 2020b). We test the weight for local model regularization in $[0.1, 0.01, 0.001]$ and report the best results.
- *FedLC*: it calibrates the logits before softmax cross-entropy according to the probability of occurrence of each class (Zhang et al., 2022a). We test the scaling factor in the calibration from 0.1 to 1 and report the best performance.

---

[1] https://www.kaggle.com/code/longx99/sound-classification/notebook

Table 6: Overall performance comparison. Accuracy is reported as $mean \pm std$ across five runs. The best baseline (excluding *FedData*) under each column is highlighted.

| | #Clients: 100 (500 samples per client on average) | | | #Clients: 500 (100 samples per client on average) | | |
| --- | --- | --- | --- | --- | --- | --- |
| | $Quantity(3)$ | $Dirichlet(0.5)$ | $Dirichlet(0.1)$ | $Quantity(3)$ | $Dirichlet(0.5)$ | $Dirichlet(0.1)$ |
| FedAvg | 43.55±0.82 | 50.36±0.89 | 28.21±1.20 | 30.25±1.33 | 32.58±1.09 | 20.46±2.15 |
| FedProx | 44.37±0.89 | 49.30±1.00 | 34.66±1.11 | 31.92±1.45 | 32.01±1.25 | 20.86±1.97 |
| FedDecorr | 44.09±0.90 | 51.27±0.93 | 30.89±1.40 | 31.12±1.57 | 33.57±1.22 | 21.34±1.59 |
| FedLC | 49.35±1.01 | 53.58±1.02 | 36.05±1.21 | 32.05±1.60 | 30.17±1.18 | 18.82±2.01 |
| FedNTD | 53.01±1.23 | 56.06±0.97 | 41.48±0.90 | 39.98±0.97 | 39.82±0.86 | 26.78±2.34 |
| FedBR | 44.58±0.73 | 51.65±1.02 | 32.11±1.45 | 31.66±1.07 | 33.08±1.12 | 20.98±2.54 |
| CCVR | 49.11±0.67 | 51.21±0.98 | 34.47±1.35 | 35.95±1.63 | 35.02±1.43 | 24.21±2.67 |
| FedGen | 46.66±2.87 | 52.89±1.09 | 33.18±1.29 | 32.32±1.21 | 34.27±1.56 | 22.56±2.89 |
| **Each client shares 10% of the data or features** | | | | | | |
| FedData | 67.60±1.33 | 72.17±1.34 | 70.34±1.68 | 54.64±1.02 | 56.47±1.22 | 55.35±1.46 |
| FedMix | 52.78±1.99 | 57.97±1.24 | 40.68±1.50 | 44.04±1.53 | 45.50±1.88 | 38.13±2.06 |
| **FLea** ($l = 5$) | 58.27±0.95 | 59.63±1.28 | 43.65±1.47 | 47.03±1.01 | 48.86±1.43 | 44.40±1.23 |

Table 7: Overall performance comparison for audio classification. Accuracy is reported as $mean \pm std$ across five runs. The best baseline (excluding *FedData*) under each column is highlighted.

| | #Clients: 70 (100 samples per client on average) | | | #Clients: 140 (50 samples per client on average) | | |
| --- | --- | --- | --- | --- | --- | --- |
| | $Quantity(3)$ | $Dirichlet(0.5)$ | $Dirichlet(0.1)$ | $Quantity(3)$ | $Dirichlet(0.5)$ | $Dirichlet(0.1)$ |
| FedAvg | 43.69±0.56 | 46.77±0.87 | 34.59±2.64 | 39.35±0.60 | 43.98±0.89 | 31.21±1.62 |
| FedProx | 38.45±0.48 | 39.58±1.02 | 34.81±0.46 | 39.05±0.56 | 42.21±0.76 | 32.85±1.22 |
| FedDecorr | 45.01±0.57 | 46.77±0.65 | 35.87±1.03 | 39.67±0.58 | 44.23±0.95 | 33.67±1.34 |
| FedLC | 50.98±0.49 | 50.11±0.83 | 37.05±0.87 | 44.33±0.79 | 45.15±0.80 | 39.87±1.04 |
| FedNTD | 44.80±0.45 | 51.09±0.97 | 36.53±0.99 | 42.21±0.63 | 48.63±0.78 | 40.15±1.22 |
| FedBR | 44.05±0.63 | 47.58±0.90 | 36.15±1.17 | 41.15±0.70 | 44.37±0.82 | 34.89±1.36 |
| CCVR | 47.12±0.72 | 49.26±0.92 | 39.62±1.20 | 44.05±0.87 | 46.68±0.83 | 36.80±1.37 |
| FedGen | 45.20±0.89 | 48.33±1.12 | 38.27±1.44 | 40.89±0.72 | 44.54±0.81 | 35.78±1.40 |
| **Each client shares 10% of the data or features** | | | | | | |
| FedData | 62.83±1.25 | 64.45±0.76 | 61.11±0.98 | 60.31±0.82 | 60.48±0.91 | 59.67±1.55 |
| FedMix | 51.56±0.59 | 54.18±0.62 | 43.35±0.72 | 46.55±0.81 | 50.00±0.92 | 42.27±1.15 |
| **FLea** ($l = 2$) | 57.73±0.51 | 59.22±0.78 | 45.94±0.77 | 54.35±0.80 | 55.68±0.87 | 45.05±1.32 |

- **FedDecorr**: This method applies a regularization term during local training that encourages different dimensions of the low-dimensional features to be uncorrelated (Shi et al., 2022). We adapt the official implementation[2] and suggested hyper-parameter in the source paper. We found that this method can only outperform *FedAvg* with fewer than 10 client for CIFAR10.

- *FedNTD*: It prevents the local model drift by distilling knowledge from the global model (Lee et al., 2022). We use the default distilling weights from the original paper as the setting are similar[3].

- **FedBR** (Guo et al., 2023): this approach leverage 32 mini-batch data averages without class labels as data augmentation. A min-max algorithm is designed, where the max step aims to make local features for all classes more distinguishable while the min step enforces the local classifier to predict uniform probabilities for the global data averages. We adapt the official implementation[4] in our framework.

- *CCVR*: It collects a global feature set before the final fully connected linear of the converged global model, i.e., the model trained via *FedAvg*, to calibrate the classifier on the server (Luo et al., 2021). For a fair comparison, we use the same amount of features as our method for this baseline, and we fit the model using the features instead of distributions as used in (Luo et al., 2021). This allows us to report the optimal performance of *CCVR*.

- **FedGen**: It is a method that trains a data generator using the global model as the discriminator to create synthetic data for local training (Liu et al., 2022). The generator outputs $\hat{x}_i$ with input $(y_i, z_i)$ where $z_i$ is a sample for Normal distribution. The generator is a convolutional neural

---

[2] https://github.com/bytedance/FedDecorr
[3] https://github.com/Lee-Gihun/FedNTD.git
[4] https://github.com/lins-lab/fedbr

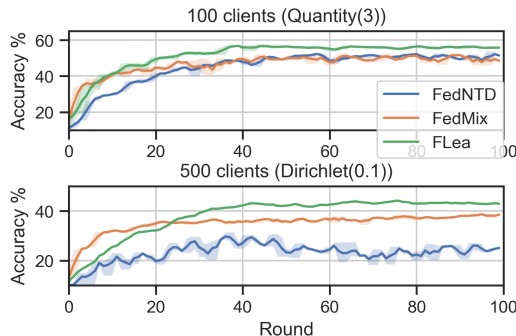

Figure 11: Accuracy of the global model for CIFAR10.

network consisting of four *ConvTranspose2d* layers to upsample feature maps. We train the first 30 rounds by normal *FedAvg* and after 30 rounds, we use the global model as the discriminator to distinguish with the generated data $\hat{x}_i$ is real or not.

- **FedData**: In this baseline, we assume the server waits until all the clients have shared $10\%$ of their local data in the beginning round. The gathered data will be sent to clients to mix with local data for model training.

- **FedMix**: Similar with *FedData*, we assume the server waits until all the clients have shared their data averages.we use a mini-batch of $10$ to aggregate the samples. Different from *FedBR* The gathered data will be sent to clients, combing with local data based on the Beta distribution.

## C.4 EXTENDED RESULTS

Tables 6 and 7 for additional results corresponding to different local data sizes, supplementing the information presented in Table 2.

***Results for CIFAR10 classification.*** The analysis for Table 6 is elaborated on as follows. Among loss-based approaches, *FedNTD* is the best, showing a strong ability to handle data scarcity and label skew (about $10\%$ improvement from *FedAvg*), while other loss-based FL methods present marginal gain from *FedAvg*. The outstanding performance of *FedNTD* are mainly attributed to its knowledge distillation component, mitigating the local over-fitting as well as model bias. *FLea* further improves *FedNTD* by $3 \sim 5\%$ when the average local data size is $500$, and the superiority of *FLea* is more remarkable with increasing level of data scarcity, e.g., when the average local data size reduces to $100$, the performance gain reaches $17.6\%$ (*Dirichlet(0.1)* group).

For data augmentation-based baselines, *FedMix* performs the best and for most of the cases, it is the SOTA baseline excluding *FedData*. When sharing the same proportion of global proxies (*FedMix* shares data averages while *FLea* shares features), *FLea* outperforms *FedMix* by $2 \sim 6\%$ across all experiments. We report the performance of *FedData* as an Oracle. It is plausible that *FLea* cannot beat *FedData* given *FedData* shares raw data with privacy protection.

*FLea* also presents more stable performance compared to *FedNTD* and *FedMix*. As shown in Figure 11, *FLea* converges after $40$ communication rounds, with notably higher averaged accuracy and smaller variance compared to the other two best baselines. We also demonstrate each component in *FLea* yields independent contribution to the overall performance in Appendix C Sec. C.5.

***Results for UrbanSound8K classification.*** Similarly to the performance for audio classification, *FLea* consistently achieve the best accuracy across different settings. Given that the total size of UrbanSound8K is smaller than CIFAR10, this audio classification has more sever data scarcity problem globally and locally. This explains why *FedMix* is the best baseline uniformly for this task. Nevertheless, *FLea* outperforms *FedMix* by $2.59\% \sim 7.80\%$.

Table 8: More results for FLea ($\alpha = 10\%$, $|K| = 100$).

| | $Quantity(3)$ | $Dirichlet(0.5)$ | $Dirichlet(0.1)$ |
|---|---|---|---|
| FedMix | 44.04±1.53 | 45.50±1.88 | 38.13±2.06 |
| FLea ($l = 5$) | 47.03±1.01 | 48.86±1.43 | 44.40±1.23 |
| FLea ($l = 5$, $\lambda = 0.5$) | 45.87±1.23 | 46.91±1.22 | 42.01±1.14 |
| FLea ($l = 5$, $\lambda_1 = 0$) | 45.16±1.06 | 46.89±1.42 | 40.98±1.09 |
| FLea ($l = 1$) | 49.67±1.12 | 50.23±1.35 | 46.17±1.30 |
| FLea ($l = 9$) | 44.05±1.11 | 44.87±1.56 | 40.19±1.27 |

Table 9: Supplemental results of CCVR ($\alpha = 10\%$, $|K| = 100$), where global features come from the later layer in different blocks of MobileNet_V2.

| | Block18 | Block17 | Block13 | Block9 | Block5 | Block1 | Raw | Our *FLea* |
|---|---|---|---|---|---|---|---|---|
| Quantity(3) | 49.11±0.67 | 50.96±0.75 | 51.24±0.68 | 52.18±0.89 | 51.77±0.73 | 51.50±0.78 | 51.53±0.81 | 58.27±0.95 |
| Dirichlet(0.5) | 51.21±0.98 | 51.98±1.03 | 52.78±1.15 | 53.04±1.12 | 53.25±0.99 | 53.14±1.07 | 52.85±0.99 | 59.63±1.28 |

## C.5 ABLATION STUDY FOR FLEA

As introduced in Sec. 4, our *FLea* leverages feature augmentation by combing local and global features according to the weights following a Beta distribution. Now we give the results to demonstrate the advantage of introducing randomness to improve the model generalization: we use fixed $\lambda$ instead of sampling it from $Beta(2, 2)$ (refer to FLea ($l = 5$, $\lambda = 0.5$) group in Table 8). We also give the results when removing $\mathcal{L}_{dis}$ from the training loss (refer to FLea ($l = 5$, $\lambda_1 = 0$) group in Table 8). It is evident that our complete version of *FLea* always performs the best.

We also discuss the impact of layer $l$, from which layer the features are extracted. It is a trade-off between between privacy protection and the utilization of features. A smaller $l$ indicates the features are closer to the raw data while the privacy vulnerability increases. In Sec. 5.3, we have demonstrated that $l = 5$ with the de-correlation loss can well defend against privacy attacks in our simulations. In Table 8, we also show that sharing features from $l = 1$ can enhance *FLea* while from $l = 9$ can lead to a slight performance decline. For real-world applications (beyond CIFAR10), we choose $l$ according to the specific performance and privacy requirements. It is also worth mentioning that, as *FLea* is designed to leverage the latest feature buffer, $l$ won't necessarily to be fixed. On the contrary, $l$ can be dynamically altered during training based on the performance and privacy requirements.

## C.6 REFLECTIONS FOR BASELINES

**More results for CCVR.** We evaluated the baseline *CCVR* by using features from different layers to calibrate the global model, and the performance is reported in Table 9. Those results clearly suggest that leveraging features from shallower layers does not lead to further performance improvements. This suggests that post-hoc calibration has limited capability in mitigating the local drift, which is the fundamental cause of degradation in FL on non-IID data. Our *Flea* shows an evidently stronger performance.

**More reflection for FedNTD.** From both Figure 1 and Table 6, we can see *FedNTD* is a strong baseline for both data scarcity and label skew. *FedNTD* was devised to address the non-IID setting, but we find it is also able to alleviate issues with data scarcity in the IID setting. This suggests global knowledge distilling can mitigate local over-fitting. However, as the data becomes scarce, the distillation ability declines, herein the performance gain drops. Instead of using local data for knowledge distilling, in *FLea*, we leverage the augmented features to distil the knowledge from the global model into the local model.

Table 10: Architecture of decoder of MobileNet_V2.

| Layer Index | #Input | Operator | #Output Channel | #Kernel | #Stride | #Output |
|---|---|---|---|---|---|---|
| 1 | $16 \times 32 \times 32$ (Feature) | conv2d | 32 | 1 | 1 | $32 \times 32 \times 32$ |
| 2 | $32 \times 32 \times 32$ | ConvTranspose2d | 32 | 3 | 2 | $32 \times 64 \times 64$ |
| 3 | $32 \times 64 \times 64$ | conv2d | 32 | 3 | 2 | $32 \times 32 \times 32$ |
| 4 | $32 \times 32 \times 32$ | conv2d | 3 | 1 | 1 | $3 \times 32 \times 32$ (Data) |

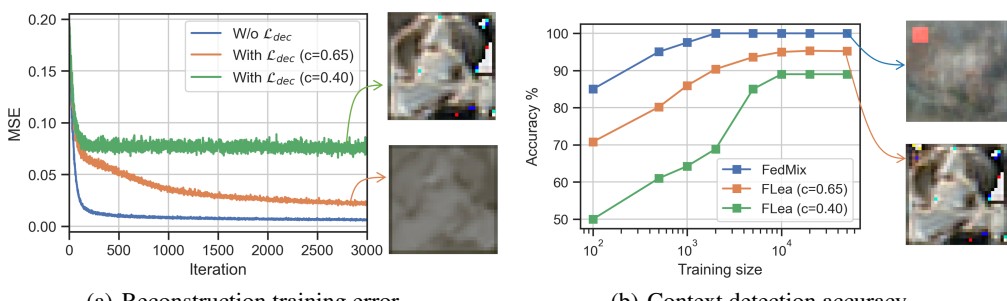

(a) Reconstruction training error.      (b) Context detection accuracy.

Figure 13: The effectiveness privacy protection. $c$ is short for the correlation in Figure 12. We show the reconstruction and context detection performance for $c = 0.65$ (the $1^{st}$ round) and $c = 0.40$ (the $10^{th}$ round).

.

## D   PRIVACY STUDY

Now we present the experimental setup for privacy attacks. We use the Quantity(3) data splits when $|K| = 100$ as an example for studying, as in other settings either the label is more skewed or the local data is more scarce, privacy attack can hardly be more effective than this setting. This is to present the attack defending for the most vulnerable case. As the correlation between the features and the data is continuously reduced (shown in Figure 12), we report the reconstruction and context detection performance for $c = 0.65$ (the $1^{st}$ round) and $c = 0.40$ (the $10^{th}$ round) for reference.

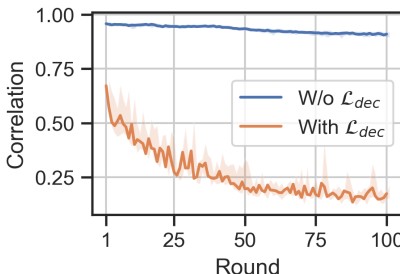

Figure 12: Correlations in each communication round.

***Data reconstruction***. We first implemented a data reconstruction attacker, following the approach described in Dosovitskiy & Brox (2016), the attacker constructed a decoder for the purpose of reconstruction. Specifically, the attacker targeted the converged global model trained using the Quantity(3) distribution, ensuring a fair comparison. The decoder architecture, designed to match the MobileNet_V2 architecture, comprised four *conv2d* layers (refer to Table 10) to reconstruct the original data from the provided features. For visualization purposes, the CIFAR10 images were cropped to a size of $32 \times 32$ pixels without any normalization. The decoder took the features extracted from the global model as input and generated a reconstructed image, which served as the basis for calculating the mean squared error (MSE).

To train the decoder, we utilized the entire CIFAR10 training set, conducting training for 20 epochs and employing a learning rate of $0.001$. This approach allowed us to evaluate the fidelity of the reconstructed data and compare it with the original input, providing insights into the effectiveness of our proposed feature interpolation method. We use the testing set and the target global model ($c = 0.65$ and $c = 0.40$) to extract features for reconstruction. Figure 13(a) shows the training MSE while the exampled images are from the testing set. For $c = 0.65$, i.e., after the first round, in Figure 12, the sensitive attributes are removed (e.g., the color of the dog). After 10 rounds when $c < 0.4$, information is further compressed and the privacy protection is enhanced. Overall, with $\mathcal{L}_{dec}$, the correlation between data and features is reduced, preventing the image from being reconstructed.

***Identifying context information***. In this attack, we assume that the attacker explicitly knows the context information and thus can generate large amounts of negative (clear data) and positive (clear data with context marker) pairs to train a context classifier (which is very challenging and unrealistic but this is for the sake of testing). Real-world attack will be far more challenging than our simulations.

The context identification attacker is interested in finding out if a given feature $f$, is from the source data with a specific context or not. We simulate the context information by adding a color square to the image (to mimic camera broken), as illustrated in Figure 5.3. We use a binary classifier consisting of four linear layers to classify the flattened features or images. To train the classifier, we add the context marker to half of the training set. To report the identification performance, we add the same marker to half of the testing set. In Figure 13(b), the identification accuracy for *FedMix* and our *FLea* are given. We measure the attacking difficulty by how many training sample the model need to achieve a certain accuracy. The results in Figure 13(b) suggest that *FLea* needs times of training sample than *FedMix* for different correlations. This demonstrates that *FLea* can better protect the context privacy.

All the above results lead to the conclusion that by reducing feature exposure and mitigating the correlation between the features and source data, *FLea* safely protect the privacy associated with feature sharing while achieving favorable performance gain in addressing the label skew and data scarcity simultaneously.

