# OpenReview forum: "FLea: Improving federated learning on scarce and label-skewed data via  privacy-preserving feature augmentation"
_ICLR.cc/2024/Conference — ICLR 2024 Conference Withdrawn Submission_

### Official Review · Reviewer_66Ba · 2023-10-28

**Soundness:** 1 poor
**Presentation:** 2 fair
**Contribution:** 2 fair
**Rating:** 3
**Confidence:** 5

**Summary:**

This paper proposes a privacy-preserving feature augmentation method to address the challenges of data scarcity and label-skewness in Federated Learning (FL). The work is well-motivated; however, there are several issues that the authors need to address to convincingly demonstrate the novelty and effectiveness of the proposed method. There are also many writing issues throughout the paper.

**Strengths:**

- The backgroud of the problem and the motivation of the work are well explained

**Weaknesses:**

Major issues:

1. The authors claim that the proposed method is novel, but many components come from the literature. For example, the authors claim that they have devised a novel feature augmentation method, but λ, a key parameter in the method, is drawn from existing literature.

2.λ1 and λ2 in Equation 6 are critical parameters, but it is unclear how they are set.

3.The paper does not discuss the impact of different model types on the proposed method or specify which layer (l-th) has been learned in the experiments. The evaluation solely relies on one AI model, MobileNet, and one dataset, CIFAR-10, which is not sufficient.

4.The statement, 'we characterize the distribution of the low-dimensional features from the penultimate layer of local and global models,' lacks justification regarding why the penultimate layer is suitable for characterizing low-dimensional features.

Other issues:
-  Section 2.2 mentions low-dimensional feature overlapping and classifier bias as main label skew problems, but it does not elaborate on low-dimensional feature overlapping or explicitly define the problem.

-The paper would benefit from a more comprehensive discussion explaining why sharing features can enhance model performance.

-Notations in B and Bf, which are crucial concepts in the work, are not defined explicitly. Although the definitions of xi, yi, and fi can be inferred, it remains unclear what y^fi represents.

-The sentence regarding (f^l_i , yi) and (fj, yf j ) should use the same notation consistently.
The use of 'λ ∼ Beta(a, a)' needs clarification. It would be helpful to explain what Beta() is and how it is relevant. The reference at the end of the sentence should specify which part of the information it pertains to.

Language problems:
-"prompting the introduction of regularization techniques to address this issue.erm, Lr, can be included"?
- suggesting that the over-fitting is [severe]
-FLea works in [an] iterative manner
-The sentence "one data batch with labels from Dk and one feature batch with labels from the received feature buffer F (t), termed by B = {(xi, yi) ∈ Dk} and Bf = {(fi, y^f i ) ∈ F (t)}, respectively (|B| = |Bf |)." is incomplete
- one for knowledge distillation [from] the global model

**Questions:**

How are the values of λ1 and λ2 in Equation 6 are set?

what is y^fi in Bf?

What are the impact of model types on the proposed method?

What is the Beta() function?

---

> ### Author Response · Authors · 2023-11-20
> **Demonstrating the novelty and effectiveness (I)**
>
> Dear reviewer:
>
> We appreciate your time reviewing our paper. We have carefully addressed all the concerns you raised and revised the submission correspondingly. Please check the responses below and the highlighted revision in the updated PDF.
>
> Response to major issues:
>
> **1. Novelty of the method:**
>
> Firstly, we would like to clarify the method regarding feature augmentation. λ was proposed for data augmentation to improve the model generalization (Zhang et al, 2018),  while **we are the first to extend it for feature augmentation in the federated learning setup**. This effectively addresses the local overfitting problem caused by data scarcity. We apologise for the wrong reference regarding the mixup in the previous PDF. Now we have corrected it to:
> Zhang, Hongyi, et al. "Mixup: Beyond Empirical Risk Minimization." International Conference on Learning Representations. 2018.
> And we revised the context to: ‘‘Inspired by the data augmentation method in the centralized setting (Zhang et al, 2018), we sample the weight $\lambda$  for each data point from a symmetrical Beta distribution (Gupta & Nadarajah, 2004):  $\lambda \sim Beta(a,a)$.   $\lambda \in [0,1]$ controls the strength of interpolation between the local and global feature pairs: A smaller $\lambda$ basically makes the generated sample closer to the local feature while a larger one pushes that to the global feature.”
>
> Although this augmentation method exists, **the novelty of our study lies in the observation that data scarcity will lead to local model over-fitting (refer to Section 3.2) and hence leveraging feature augmentation can effectively address this problem**. Data scarcity is a common issue in FL but its impact is not very understood. We provide a comprehensive analysis of this issue and propose the feature augmentation-based method FLea. As demonstrated by the extensive experiments, our method achieves a significant model accuracy boost compared to the start-of-the-art FL baselines. Moreover,  instead of sharing raw data and implementing data augmentation, we also demonstrate that feature augmentation can better trade off the feature utility and data privacy protection.  Given these results, we believe our contribution to the FL community is significant.
>
> **2. How $\lambda_1$ and $\lambda_2$  are set:**
>
> In this updated version, we supplement how we identify the hyper-parameters $\lambda_1$ and $\lambda_2$  in the loss function in *Section 5.2 Impact of pyper-parameters*. We present a heuristic search and identify the results considering the model accuracy and the data privacy simultaneously. Finally, it has been observed that the model achieves the best trade-off when $\lambda_1$ ranges from 1 to 2 (refer to Figure 7(a)) and when  $\lambda_2$ ranges from 1 to 4 (refer to Figure 7(b)). Finally, we set $\lambda_1=1$ and $\lambda_2=3$ for all experiments.
>
> **3. Insufficient model and data:**
>
>  As per our submitted code, we implemented our methods for various architectures and datasets, which we did not include before due to the limited spaces. Now, according to your suggestion, we add the results on another data modality-audio with a CNN model. The details can be found in Section 5, and Table 2 has been updated (including the results for UrbanSound8K in the right part), with extensive results and discussions presented in Appendix C.4 (page 17-18). The results show that FLea consistently outperforms the state-of-the-art baselines by 2.59~6.27%. For your convenience, we quote part of the results below:
>
> Table 2 Overall performance comparison. The local data size $|\mathcal{D}_k|$ is as small as $100$ on average.
> | Accuracy %       |     |          CIFAR10       |                | |        UrbanSound8K        |                |
> | ------ | ----------- | -------------- | -------------- | ------------ | -------------- | -------------- |
> |               | Quantity(3) | Dirichlet(0.5) | Dirichlet(0.1) |    Quantity(3)  | Dirichlet(0.5) | Dirichlet(0.1) |
> | **FedMix**|   $44.04{\pm1.53}$ |$45.50{\pm 1.88}$ |$    38.13{\pm2.06}$   |$51.56{\pm 0.59}$ |$54.18{\pm 0.62}$  |$43.35{\pm 0.72}$
> | **FLea**  |    $47.03{\pm1.01}$|$48.86{\pm1.43}$|$44.40{\pm1.23}$| $     57.73 {\pm 0.51}$ |$59.22 {\pm 0.78}$ |$45.94{\pm 0.77}$

---

> ### Author Response · Authors · 2023-11-20
> **Demonstrating the novelty and effectiveness (II)**
>
> **4. Why the penultimate layer is suitable for characterizing low-dimensional features?**
>
> Deep neural networks usually consist of multiple linear and non-linear layers, making their functioning difficult to explain. It is a common way to divide the model into two parts: the prototype extractor (from the input to the penultimate layer) and the linear classifier (the last fully connected layer) [1,2,3]. The prototypes have lower dimensionality than the features from other intermedia layers, which is more straightforward for visualisation (after dimension reduction). We follow the setup from previous work to carry out our analysis.  We have explained the suitability of this more clearly in the paper.
>
> [1]Tan, Yue, et al. "Fedproto: Federated prototype learning across heterogeneous clients." Proceedings of the AAAI Conference on Artificial Intelligence. Vol. 36. No. 8. 2022.
>
> [2]Li, Xin-Chun, et al. "Fedrs: Federated learning with restricted softmax for label distribution non-iid data." Proceedings of the 27th ACM SIGKDD Conference on Knowledge Discovery & Data Mining. 2021.
>
> [3]Guo, Yongxin, et al. "FedBR: Improving Federated Learning on Heterogeneous Data via Local Learning Bias Reduction." (2023).
>
> Response to other issues:
>
> **1.Section 2.2 does not elaborate on low-dimensional feature overlapping or explicitly define the problem:**
>
> Section 2.2 delves into insights gathered from prior research. In [3], it was discerned that owing to label skew, "local features, even for data from different classes, are too similar to one another to be accurately distinguished (cf. Figure 1(b))," thus giving rise to low-dimensional feature overlapping. As this is not the primary focus of our contribution, we have provided a concise discussion of it in  Section 2.2 in the interest of directing the reader's attention to our key insights in Sections 2.3 and 3, where we delve into the issue of data scarcity.
>
> **2. A more comprehensive discussion explaining why sharing features can enhance model performance:**
>
> Our approach enhances model performance in scenarios characterized by both data scarcity and label skew through two key aspects: 1) leveraging features from other clients, particularly those of locally absent classes, to alleviate local bias, and 2) employing feature augmentation across all classes, to enhance local model generalization. The effectiveness of the first aspect is validated by the outstanding performance FLea achieves in various non-iid settings, with a more pronounced improvement observed in the most skewed case, as illustrated in Table 2. The second aspect is demonstrated by consistently outperforming state-of-the-art methods designed for addressing label skew but not explicitly considering overfitting. We have also incorporated this summarization into the results section of the updated PDF.
>
> **3. Unclear Notations and Formulation:**
>
> $(x_i, y_i)$ denotes a data-label pair from the local data set, and $(f^{\mathcal{F}}_i,y^{\mathcal{F}}_i)$ denotes a feature-label pair from the feature buffer. $\mathcal{B}$ presents a batch of data-label pairs, i.e. $\mathcal{B}= ${$(x_i,y_i)$}, and $\mathcal{B}^f$ presents a batch of feature-label pairs, i.e. $\mathcal{B}^f=${$(f^{\mathcal{F}}_i,y^{\mathcal{F}}_i)$}.
>
> Beta distribution is a continuous probability distribution defined on the interval [0, 1]. It is characterized by two shape parameters, usually denoted as $\alpha$ and $\beta$, which determine the shape of the distribution. The probability density function (PDF) of the Beta distribution is given by,
>  $ f(\lambda; \alpha,\beta) = \frac{\lambda^{\alpha-1} (1-\lambda)^{(\beta-1)} }{N}$, where $N$ is the normalizing factor and $\lambda\in[0,1]$. In our study, we choose $\alpha=\beta=a$ and herein,   $f(\lambda)= \frac{1}{N} (\lambda (1-\lambda)) ^{\alpha-1}$, which is symmetrical. We move the reference (zhang et al. 2018) to where it is mentioned to make the explanation clear.
>
> According to your comments, we have thoroughly improved the notations and formulations in the method introduction part to improve the coherence and clarification. Please check our revised pdf.
>
> **Language problems:**
>
> We have addressed the typos and thoroughly reviewed and revised the entire submission with contributions from all authors.
>
> **Response to  Questions:**
>
> **How are the values of λ1 and λ2 in Equation 6 are set?**
>
> Please refer to our response to Major Issue 2.
>
> **what is y^fi in Bf?**
>
> It is the class label for the feature $f^{\mathcal{F}}_i$. We have improved the notations as mentioned above.
>
> **What are the impact of model types on the proposed method?**
>
> Our method is effective for various data modalities and model architectures. Please refer to our response to Major Issue 3.
>
> **What is the Beta() function?**
>
> Beta distribution is a continuous probability distribution defined on the interval [0, 1]. Please refer to our answers above.

---

### Official Review · Reviewer_BqfA · 2023-10-28

**Soundness:** 2 fair
**Presentation:** 3 good
**Contribution:** 3 good
**Rating:** 6
**Confidence:** 4

**Summary:**

Aiming to address both the scarcity and label-skewness of data simultaneously, this work proposes the sharing of features across clients  as an extension to the classical federated averaging algorithm. Specifically, features of the $\ell$-th layer are extracted and distributed across clients. During local training, the shared features and their corresponding labels are used to refine the layers subsequent to the $\ell$-th layer. To alleviate the privacy risks associated with sharing features, a decoupling loss function is introduced to reduce the correlation between the input data and its corresponding features.

**Strengths:**

i. By illustrating the overfitting problem and the gathering effect of features (as shown in Figure 2), this work effectively demonstrates the advantages of sharing data in situations of data scarcity. This supports the proposed framework for feature sharing, particularly when data sharing is restricted due to privacy concerns.

ii.Given that federated learning is inherently a privacy-preserving framework, the act of sharing features does increase the risk of information leakage comparing with FedAvg. To mitigate this, the work introduces an additional decoupling loss to strike a privacy-utility trade-off. A comprehensive study evaluating the amount of private information compromised through feature sharing is performed.

iii. Experimental results indicate that FLea consistently and significantly outperforms the baselines in scarce and label-skewed data scenario.

**Weaknesses:**

i. My primary concern is that the comparison is limited to a single dataset (CIFAR-10) and a single architecture (MobileNet). The experiments cannot sufficiently demonstrate the generalizability of Flea across different applications.

ii. The issue of data scarcity and label skew has previously been studied, particularly in Bayesian Federated Learning frameworks. For instance, reference [1] conducted experiments in the same setting. Given that Bayesian FL does not necessitate the additional communication of features, it appears to be a safer alternative to Flea. A comparison would be valuable to ascertain whether Flea can achieve superior accuracy.

iii. The first two conclusions in the Section 3.1 are not very convincing. Specifically, "The performance of FL methods decreases remarkably as data scarcity and label skew increase." This argument is concluded from "FedAvg, its accuracy decreases from 75% to 56% when |Dk| reduces from 5000 to 100 in the IID setting. When local data is sufficient (|Dk| = 5000), its accuracy drops from 75% for the IID setting to 60% for the non-IID setting." However, it is unclear why "|Dk| reduces from 5000 to 100" can be compared with changing IID to non-IID setting. Additionally, "Loss-based methods can address label skew only with sufficient local data." This argument is too strong. As mentioned in the second point, Bayesian frameworks are more amenable to data scarcity without requiring more data.

[1] Confidence-aware Personalized Federated Learning via Variational Expectation Maximization, Zhu et al., CVPR 2023.

**Questions:**

i. The plot of accuracy as a function of $\lambda_2$ can better clarify the privacy-utility trade-off.

ii. It seems that differential privacy can also be employed to obfuscate the features. Since DP is more mathematically rigorous and widely accepted, it may be more interesting to replace $\ell_{dec}$ with DP.

---

> ### Author Response · Authors · 2023-11-20
> **Results for another dataset and model (I)**
>
> Dear reviewer,
>
> We thank you for your positive comments. We made earnest efforts to address your concerns, resulting in a significant improvement in our submission. A summary of our responses is provided below.
>
> **W-i. The comparison is limited:**
>
> As you can check from our submitted code, we implemented our methods for various architectures and datasets, which we did not include due to the limited spaces. Now, according to your suggestion, we supplement the results on another data modality - audio with another CNN model. The details can be found in Section 5, and Table 2 has been updated. The extensive results and discussion are also available in Appendix C.4 (page 17-18). We include part of the results below. The results show that FLea consistently outperforms the state-of-the-art baseline by 2.59% to 6.27% across the two different tasks. Please also check the details from the pdf.
>
> Table 2 Overall performance comparison. The local data size $|\mathcal{D}_k|$ is as small as $100$ on average.
> | Accuracy %       |     |          CIFAR10       |                | |        UrbanSound8K        |                |
> | ------ | ----------- | -------------- | -------------- | ------------ | -------------- | -------------- |
> |               | Quantity(3) | Dirichlet(0.5) | Dirichlet(0.1) |    Quantity(3)  | Dirichlet(0.5) | Dirichlet(0.1) |
> | **FedMix**|   $44.04{\pm1.53}$ |$45.50{\pm 1.88}$ |$    38.13{\pm2.06}$   |$51.56{\pm 0.59}$ |$54.18{\pm 0.62}$  |$43.35{\pm 0.72}$
> | **FLea**  |    $47.03{\pm1.01}$|$48.86{\pm1.43}$|$44.40{\pm1.23}$| $     57.73 {\pm 0.51}$ |$59.22 {\pm 0.78}$ |$45.94{\pm 0.77}$
>
>
>
> **W-ii. A comparison to [1]:**
>
> [1] is relevant to our study but the primary goal is different. We aim to enhance the global model (GM) for global distribution while this paper adapts the local model in the local distribution via personalised federated learning, although this also benefits the global model. As per the results shown in [1], in the non-iid and data scarcity scenarios, the proposed GM achieved a very marginal performance gain compared to FedAvg. Specifically, for CIFAR10 with 100 clients, the GM of the proposed method in Table 1 yields an accuracy of 60.1%, against 59.4% of FeaAvg. On the contrary, on the same dataset and with the same number of clients, our method FLea outperforms FedAvg by 9.27~15.44%, as shown in Table 6 (page 17). This demonstrates our advantage in achieving a better global model in the scarce and non-iid setting.
>
> We thank the reviewer for pointing out this paper and we have included it in our related work, as blew.
> "The above-mentioned methods are developed to prevent the global model from diverging under label
> skew, while another way to cope with this issue is to learn a personalized model per client, with the
> goal of enhancing performance within their local data distribution (Kulkarni et al., 2020; Tan et al.,
> 2022a; Kotelevskii et al., 2022). Recently, personalized FL methods based on variational Bayesian
> inference have shown promising results, supported by theoretical guarantees (T Dinh et al., 2020;
> Zhang et al., 2022b; Zhu et al., 2023). However, these methods still face challenges in learning an
> optimal global model."
>
>
> **W-iii. The first two conclusions in Section 3.1 are not very convincing:**
>
> We agree with this opinion and we have revised the writing in the updated PDF to make the conclusions more precise.
> For your question ‘it is unclear why "|Dk| reduces from 5000 to 100" can be compared with changing IID to non-IID setting’: we don’t compare the reduction in data size with the changing to non-IID setting, instead, we show that both the reduction in the amount of data and the skewness of the class can lead to performance degradation.
> For your comment ‘Loss-based methods can address label skew only with sufficient local data, This argument is too strong’:  we revised it to ‘The compared loss-based methods can address label skew only with sufficient local data’.

---

> ### Author Response · Authors · 2023-11-20
> **Results for another dataset and model (II)**
>
> **Q-i. The plot of accuracy as a function of $\lambda_2$:**
>
>  We have now added this in Section 5.2 Impact of hyper-parameters. As shown in Figure 7(c),  with the increasing of $\lambda_2$,  the accuracy drops only marginally, while privacy is better protected. We formulate privacy protection by the reduced correlation between the feature and the original data ($1-\bar{c}$) . As demonstrated in Section 5.3, reducing the correlation can effectively prevent the data from being reconstructed from the features.  Thus, the privacy protection and the utility can be balanced with $\lambda_2$ ranging from 2 to 4.
>
> **Q-ii. To replace ℓ_dec with DP:**
>
> FLea introduces feature sharing in addition to model parameter sharing. Existing methods have studied privacy concerns raised by sharing model parameters, for example, using differentially private SGD [3]. These methods can be trivially combined with Flea (for example, using DP-SGD instead of SGD in the local training) and thus this is not discussed in the paper. The less obvious case in this context is the privacy risk posed by the feature-sharing which we empirically analyse in this paper.
>
> Moreover, defining differential privacy in this context is challenging (and thus is a separate research in our opinion). Suppose, we want to protect the presence of a particular sample in the local training data. A reasonable definition of DP could be the following: if the adversary finds the ‘exact’ feature corresponding to the target data sample (using the global model), no shared feature should be nearby. Thus the ‘value’ of the feature vectors needs to be protected. Here the definition of ‘nearby’ depends on the model, dataset, training process and stage, distribution of the data in the clients, etc. Further, making this independent of the strength of the adversary (the core promise of DP) is challenging as the feature needs to be translated to either data point or to presence/absence using certain computation methods. While such data obfuscation has been studied in plausible deniability [4], obfuscating high dimensional real-valued vectors that may transmit multiple times is more challenging.
>
> [2]Yang, Zhiqin, et al. "FedFed: Feature Distillation against Data Heterogeneity in Federated Learning." Thirty-seventh Conference on Neural Information Processing Systems. 2023.
>
> [3]Abadi, Martin, et al. "Deep learning with differential privacy." Proceedings of the 2016 ACM SIGSAC conference on computer and communications security. 2016.
>
> [4] Bindschaedler, Vincent, Reza Shokri, and Carl A. Gunter. "Plausible deniability for privacy-preserving data synthesis." arXiv preprint arXiv:1708.07975 (2017).
>
> For your convenience, we quote the revision in *Section Conclusion* as follows,
>
> 'Limitations. In reality, because of extra feature sharing, FLea can introduce some extra overheads like communication and storage. We leave the improvement of the efficiency for real-world applications for future work. To enhance privacy, FLea can be trivially combined with methods protecting model parameters (such as DP-SGD[3]), however, improving the statistical privacy risks posed by feature sharing is hard due to the challenges originating from the stochasticity in modelling, data distribution in the clients, the high dimensionality of real-valued features, etc. We thus leave this as a future work. We anticipate that our work will inspire further investigations to comprehensively study feature sharing in the low data regime.'

---

> > ### Comment · Reviewer_BqfA · 2023-11-20
> > **Official Comment by Reviewer**
> >
> > Thank the authors for answering my questions and providing new results. My concerns have been addressed. I would like to maintain my score and recommend acceptance.

---

### Official Review · Reviewer_mFBA · 2023-10-31

**Soundness:** 2 fair
**Presentation:** 2 fair
**Contribution:** 2 fair
**Rating:** 5
**Confidence:** 3

**Summary:**

The article conducts an in-depth analysis of prior research in the field of federated learning, culminating in a comprehensive exploration of the issues stemming from label-skew and data-skew, and their consequential impacts on overfitting and model bias. To address these issues, the article undertakes a rigorous theoretical examination, and subsequently introduces the Flea framework as a proposed solution.

**Strengths:**

The paper comprehensively examines the phenomena of Label-Skew and Data -Skew in the context of federated learning, presenting methods that have demonstrated a significant enhancement in model performance while preserving privacy

**Weaknesses:**

1. The idea of broadcasting features in the paper appears to make a relatively modest contribution. The proposed under-explored scenarios of Label-skewed and Data-skewed resemble another expression of Non-iid data, which is not an under-explored scenario itself. It is unclear whether the authors can provide an explanation for the distinctions between these scenarios and Non-iid data.

2. The concept of overfitting and Client-Drift resulting from Label-skewed and Data-skewed scenarios seems consistent and has been extensively investigated in prior research.

3. Authors says this feature-sharing method is privacy-preserving, but it will leak more privacy compared to other algorithm without feature-sharing

4. The experiments in the paper exhibit depth but lack breadth.
   a) Lack of domain generalization: The paper's experimental dataset is limited to CIFAR10, without inclusion of other datasets.
   b) Lack of model generalization: The experimental investigation in the paper is limited to a single MobileNet_V2 model, and the discussion regarding the selection of feature layer 'l' is somewhat vague.
   c) The paper employs feature broadcasting but lacks a comparative study of communication costs, and the two communication rounds increase overhead.
   d) Observing that three loss terms act on $\theta_k$ in Eq 6, the ablation experiments are not presented in the main text. It is recommended that the authors consider reducing the discussion of Label-skewed and Data-skewed scenarios to make the core arguments more concise.

**Questions:**

This paper focuses on scarce and label-skewed data in federated learning. This is a kind of non-iid case, and a lot of papers study the non-iid and heterogenous scenarios in FL, why authors say this is the first research?

---

> ### Author Response · Authors · 2023-11-20
> **Justifying the scenarios and adding more experimental results (I)**
>
> Dear reviewer,
>
> Firstly of all, we would like to highlight that our focus is on studying Label-Skew and **Data-Scarcity** in FL, rather than **Data-Skew** which you mentioned. *Data scarcity*, as we define it, refers to scenarios where the amount of data residing on clients is limited. One example is the cross-device setting where each device collects only a few samples. In such situations, we have discovered that local models are prone to overfitting, which consequently hinders the global model from reaching optimality.  Our definitions and discussions can be found in *Section 1 Introduction* and *Section 3 Limitations of Previous Work and Insights*. It is also worth noting the positive feedback from *Reviewer vPSz*, who praised our paper for proposing feature-level augmentation for FL in a heterogeneous and low data regime. Additionally, both *Reviewer BqfA* and *66Ba* regard addressing the scarcity problem in FL as a strength of our paper.
> Regarding the weaknesses you pointed out, we have addressed them one by one and our responses are summarized as follows. Corresponding revisions have been made in the updated pdf, highlighted in blue.
>
> **W1. The distinctions between these scenarios and non-iid data:**
>
> We would like to clarify our definition of label skew and data scarcity. **Label skew** is a kind of non-IID where local class distribution is skewed. **Data scarcity** refers to the scenario where local data is very limited. We didn’t define **data skew**, but we suppose you mean the data quantity disparity, i.e.,  the amount of local data across clients varies.
>
> Label skew and data quantity disparity usually happen together. One example is that the widely adapted data split method in FL literature employs Dirichlet distribution, resulting in label and quantity skew simultaneously [1]. Specifically, if we distribute the training set of CIAFR10 (50,000 samples) to 10 clients using a Dirichlet distribution parameterized by 0.1, these clients will present different class distributions, and the total number of local samples ranges from 2853 to 8199. This is the commonly explored non-IID setting. In this paper, we further explore scarce non-IID data. Continuing with the abovementioned CIFAR10 examples, we now split the data into 100 and 500 clients, respectively. As a result,  the number of samples per client reduces significantly: the median number drops from 5685 to 90 when the number of clients increases from 10 to 500, as shown in Figure 10(a). This is the realistic scenario that we are interested in. It is also worth mentioning that data scarcity is independent of label skew and it can happen in the IID scenario. As shown in Figure 10(b), the local data covers 10 classes uniformly, but the data scarcity problem becomes severe when the number of clients increases.
>
> Overall, the label skew and quantity disparity have been studied, but to the best of our knowledge, we are the first to deeply look into the data scarcity issue in FL. To make the data scarcity senior more clever, we have supplied Figure 10 and the discussion in Appendix C.1.
>
> [1] Zhang, Jie, et al. "Federated learning with label distribution skew via logits calibration." International Conference on Machine Learning. PMLR, 2022.

---

> ### Author Response · Authors · 2023-11-20
> **Justifying the scenarios and adding more experimental results (II)**
>
> **W2. The concept of overfitting and client-drift:**
>
>  They are different in our paper and overfitting caused by data scarcity in FL is under-explored.  Overfitting is a common issue in deep learning, and it occurs when a model learns the training data very well but it can hardly generalize to new, unseen data [2]. One primary reason leading to overfitting is that the amount of training data is limited, and thus the model might memorize the examples rather than learning the underlying patterns. A more detailed illustration can be found in Section 3.2 where we use an IID setting to explain overfittting. The concept of *client drift* was proposed to illustrate the phenomenon in federated learning [3,4]. With non-IID data, the local model is optimised by local distribution, which leads to a sub-optimal global model after aggregation.  Here, the local model could be overfitted (biased) to local distribution and thus cannot generalize to the global distribution. On the contrary, in our studied setting, because of the limited local data (small in data size), the model can be thoroughly overfitted and cannot generalize to the unseen data from the local distribution. Therefore, our identified overfitting and client drift are orthogonal in FL, rather than overlapping.  To make this clearer, we have revised the Introduction when we introduce the concepts in the updated submission: ‘...data scarcity, which refers to the situation where all clients have a limited number of samples.” and ‘‘Label skew alone can lead to model bias: local models are over-fitted by the local distribution and struggle to generalize to the global distribution. This is known as client drift, which consequently leads to a sub-optimal global model”.
>
> [2] Brownlee, Jason. Better deep learning: train faster, reduce overfitting, and make better predictions. Machine Learning Mastery, 2018.
> [3]Zhao, Yue, et al. "Federated learning with non-iid data." arXiv preprint arXiv:1806.00582 (2018).
>
> [4]Karimireddy, Sai Praneeth, et al. "Scaffold: Stochastic controlled averaging for federated learning." International conference on machine learning. PMLR, 2020.
>
> **W3. Authors say this feature-sharing method is privacy-preserving, but it will leak more privacy compared to other algorithms without feature-sharing:**
>
> We agree that our method will have more privacy risks than other FL counterparts without feature-sharing. However, FLea demonstrates superior performance in model accuracy as presented in Section 5.2 Results. For instance, FLea outperforms the state-of-the-art non-feature sharing method FedNTD by 17.6% when there are 500 clients with an average data size of 100 in CIFAR10 classification. Our study finds that with extremely scarce data, sharing some global information along with the model parameters is necessary to improve the performance of FL.
>
> We want to clarify that our analysis on privacy preservation mainly is compared to sharing raw data samples (FedData) or the average of data samples (FedMix) across clients. We demonstrate that compared to those two baselines which require data exchanging globally in the beginning, in our method feature sharing only occurs within a small number of participating clients and thus can significantly reduce the data exposure (refer to Figure 8). Moreover, since we have obfuscated the features by reducing the correlation between the feature and the raw data, we also demonstrate that our method can defend against data reconstruction and context information identification (refer to Figure 13). We have made this clearer in Section 5.3 in the updated submission.

---

> ### Author Response · Authors · 2023-11-20
> **Justifying the scenarios and adding more experimental results (III)**
>
> **W4. The experiments in the paper exhibit depth but lack breadth:**
>
> **For a) and b) regarding experiments**, we have now supplemented the results for audio classification, using the UrbanSound8K data and a different CNN architecture. Table 2 has been updated to include the new results, with extensive results and discussions presented in Appendix C.4 (pages 17-18). The results demonstrate that FLea consistently outperforms the start-of-the-art baseline by 2.59%~6.27% across the two tasks.  We include the results for FLea compared to the SOTA FedMix blew,
>
> Table 2 Overall performance comparison. The local data size $|\mathcal{D}_k|$ is as small as $100$ on average.
> | Accuracy %       |     |          CIFAR10       |                | |        UrbanSound8K        |                |
> | ------ | ----------- | -------------- | -------------- | ------------ | -------------- | -------------- |
> |               | Quantity(3) | Dirichlet(0.5) | Dirichlet(0.1) |    Quantity(3)  | Dirichlet(0.5) | Dirichlet(0.1) |
> | **FedMix**|   $44.04{\pm1.53}$ |$45.50{\pm 1.88}$ |$    38.13{\pm2.06}$   |$51.56{\pm 0.59}$ |$54.18{\pm 0.62}$  |$43.35{\pm 0.72}$
> | **FLea**  |    $47.03{\pm1.01}$|$48.86{\pm1.43}$|$44.40{\pm1.23}$| $     57.73 {\pm 0.51}$ |$59.22 {\pm 0.78}$ |$45.94{\pm 0.77}$
>
>
> **For c) regarding the communication cost**, we acknowledge that FLea achieves superior model accuracy at the cost of more communication overheads and computation burden compared with non-augmentation methods, but FLea is more advanced in the convergence speed. To validate this, we have now added model learning speed analysis in Figure 5.  It is evident that FLea consistently requires fewer communication rounds to attain a target model accuracy.  This suggests that FLea proves advantageous in scenarios where extensive communication with a large number of clients is not always feasible.
>
> Moreover, compared to the augmentation-based methods like FedData and FedMix, we also demonstrate that with similar communication and computation costs, FLea yields significant accuracy gain (refer to Section 5.2) and effectively reduces the privacy risk (refer to Section 5.3).
>
> We have now added the discussion to Section 5.2 and Section 6 Conclusions.
>
> **For d) regarding the organization of the paper,** following your suggestion, we have now shortened the discussion for related work and given more space for experimental settings and results. More concretely, we have added a hyper-parameter study in Section 5.2 with a deep analysis of the weight in the loss function. As shown in  Figure 7, we demonstrate how we find the weight in the loss function to achieve the best performance improvement and privacy protection trade-off.
>
>
> **Questions why authors say this is the first research on non-iid FL?**
>
> **Response:** Given our answers to W1 and W2 as above, to the best of our knowledge, we are the first to decouple data scarcity from label skew in the non-iid FL. We found that label skew leads to local model bias while data scarcity leads to local overfitting. Label skew has gained extensive attention but data scarcity is relatively under-explored. One piece of evidence is that most related publications split CIFAR10 into 10~100 clients, while we experiment on 500 clients, which leads to significant small local datasets.
> Addressing these two problems is equally important for real-world FL applications and our study pushes the envelope. To make this clearer, we have rephrased our first contribution in the INTRODUCTION to 'The first study on a common but under-explored scenario in FL, where all the clients possess limited and highly label-skewed data. We find that model over-fitting caused by data scarcity is under-looked by existing methods.'

---

### Official Review · Reviewer_vPSz · 2023-11-01

**Soundness:** 2 fair
**Presentation:** 2 fair
**Contribution:** 2 fair
**Rating:** 5
**Confidence:** 3

**Summary:**

This paper proposes feature level augmentation for FL on heterogeneous and low data regime. Alongside the model parameters server sends a feature buffer (that includes feature, label pair ) to clients as side information. The authors show that this side information can be used to increase the testing performance in label-skew and data scarce settings.

**Strengths:**

- As far as I'm aware sending feature, label pairs is a new idea.
- The presentation of the algorithm through Figure 4 and the visuals presented in the first 3 sections are very helpful.
- There is a decent increase in the performance compared to non-augmentation methods.

**Weaknesses:**

- More details needed on feature buffer, until very late it is not obvious what is the number of pairs and how it is collected.
- The scale of the experiments (number of samples and clients) is good, but more datasets are needed for evaluation.
- In section 3.1 authors report some performance changes due to data scarcity etc. but the setting is not clear (e.g. what is the communication frequency/local iterations).
- The parameter $a$ is very critical, yet its resulted sensitivity is not thoroughly analyzed in experiments. Also introducing such a parameter is not ideal for FL settings.
- Communication and computation burden due to the added feature buffer is not examined adequately.
- By choosing $\lambda_2$ authors adjust how private the algorithm is but since there is no rigorous privacy utility tradeoff; it is hard to characterize the choice of $\lambda_2$. I think this lack of rigor is undesirable in FL.

**Questions:**

Please address the weaknesses above.

---

> ### Author Response · Authors · 2023-11-20
> **Response to the experimental setup and parameter tuning (I)**
>
> Dear reviewer,
>
> Thanks for your review. We would like to respond to your concerns:
>
> **W1.  More details on feature buffer:**
>
> Feature buffer stores the feature-label pairs from the previous communication round. As shown in Figure 4, in each round, this buffer gathers the feature-target pairs from several selected clients and each client contributes the features from a fraction of its local data. To be more specific, for each selected client n, we first randomly hold out a fraction ($\alpha$) of its data $D_k$, and extract features from the held-out data. Finally, all the extracted feature-label pairs will be sent to the server to construct the feature buffer. To make this clearer, we improve the introduction of our method in Section 4 by introducing it earlier and explicitly.
>
> The number of pairs in the buffer is not a fixed value, it depends on the number of selected clients and their data size.  In our experiments,  10% of the clients participate in each round and features from 10% of the local data of the selected clients will be sent to the buffer. For example, in the CIFAR10 experiments where we set up 500 clients ($|D_n|$ is about 100 and the number of shared features is about 10) and 50 of them are selected in each round, the size of the feature buffer will be about $10\times50=500$.  Keeping the fraction small can reduce the communication burden and privacy risk. In addition,  we also explore the performance for sharing different fractions, as shown in Figure 6. It demonstrates that $\alpha=$10% is good enough to boost the performance.
>
> **W2. More datasets:**
>
> We had implemented and tested our methods for various architectures and datasets previously (as can be checked from the submitted code), but did not include them due to the limited spaces. Now, following your suggestion, we supply the results on another data modality-*audio* with a CNN model in the main paper. Section 5 and Table 2 have been updated (including the results for UrbanSound8K in the right part), with extended results presented in Appendix C.4 (page17-18). The results show that FLea consistently outperforms the start-of-the-art by 2.59%~6.27%. For your convenience, we quote part of the results below:
>
> Table 2 Overall performance comparison. The local data size $|\mathcal{D}_k|$ is as small as $100$ on average.
> | Accuracy %       |     |          CIFAR10       |                | |        UrbanSound8K        |                |
> | ------ | ----------- | -------------- | -------------- | ------------ | -------------- | -------------- |
> |               | Quantity(3) | Dirichlet(0.5) | Dirichlet(0.1) |    Quantity(3)  | Dirichlet(0.5) | Dirichlet(0.1) |
> | **FedMix**|   $44.04{\pm1.53}$ |$45.50{\pm 1.88}$ |$    38.13{\pm2.06}$   |$51.56{\pm 0.59}$ |$54.18{\pm 0.62}$  |$43.35{\pm 0.72}$
> | **FLea**  |    $47.03{\pm1.01}$|$48.86{\pm1.43}$|$44.40{\pm1.23}$| $     57.73 {\pm 0.51}$ |$59.22 {\pm 0.78}$ |$45.94{\pm 0.77}$
>
>
> **W3.  Setting for Sec 3.1:**
>
>  Section 3.1 involves IID and non-IID data splits with different local data sizes and local class distribution. To make it clearer, we have added a group index (a,b,c,d,e,f) in Figure 1 to ease the presentation and supplemented the experimental setup in Appendix A (page 13). Briefly, in (a,b,c) local data contains all 10 classes uniformly while in (d,e,f)  local data only covers randomly selected 3 of the 10 classes. For the local data size, (a)>(b)>(c) and (d )>(e)>(f). Please check our updated submission for more details.
>
> **W4. The sensitivity of parameter $a$:**
>
> The performance of FLea is not sensitive to the choice of a. As formulated in Eq.(2), for each data point, we sample the mixup weight λ from a beta distribution, i.e., $\lambda$\~Beta(a,a) with parameter a controlling the sharpness of the beta distribution. Λ ranges from 0 to 1: A smaller $\lambda$ basically makes the generated sample closer to the local feature while a larger one pushes that to the global feature. The Beta distribution is originally controlled by two parameters  $\lambda$\~Beta($\alpha$,$\beta$) [1], and we use  $\alpha=\beta=a$ to ensure that the distribution is symmetrical and expectation of $\lambda$ is 0.5, so that the global features and local features have equal opportunities to be used for training. We have also added an empirical study in Section 5.2 to illustrate the model performance with varying a. Figure 7(c) shows the pdf for Beta distribution with the corresponding accuracy, It is evident that the results are not sensitive to $a (a>1)$.
>
> [1] Gupta, Arjun K., and Saralees Nadarajah, eds. Handbook of beta distribution and its applications. CRC press, 2004.

---

> ### Author Response · Authors · 2023-11-20
> **Response to the experimental setup and parameter tuning (II)**
>
> **W5. Communication and computation burden:**
>
> Firstly, we would like to acknowledge that FLea yields more communication overheads and computation burden compared with non-augmentation methods, as additional features (which can often be as large as the model) need to be transferred and used for training.  We leave a detailed investigation for the future as the sizes can be optimized in various ways like sharing features every several rounds, compressing the feature vectors, caching previously shared features at the clients, etc., each of which can constitute its own line of investigation.
>
>
> However, it is worth noting that our study focuses on showing the promise of sharing features against sharing other data augmentations like raw data or the average of raw data. We show that with similar communication and computation costs, FLea significantly outperforms FedMix (refer to Section 5.2) and effectively reduces the privacy risk of FedData and FedMix (refer to Section 5.3).
>
> Moreover, compared to the non-augmentation methods, we also demonstrate that although requiring more communication overheads, FLea not only achieves significantly higher final accuracy but also exhibits faster learning. For this purpose, we supplement a convergence speed analysis in Figure 5.  From the figure, it is evident that FLea consistently requires fewer communication rounds to attain a target model accuracy.  This suggests that FLea proves advantageous in scenarios where extensive communication with a large number of clients is not always feasible.
> We have now added the discussion to Section 5.2 and Section 6 Conclusions.
>
>
>
> **W6. The choice of $\lambda_2$:**
>
> We have now added a privacy protection and feature utility trade-off analysis in Figure 6(c). We formulate the privacy protection by the reduced correlation between the feature and the original data ($1-\bar{c}$) and the utility by the final model accuracy. As $\lambda_2$ increases, $1-\bar{c}$ increases correspondingly while the accuracy presents an acceptable drop. Since reducing the correlation can effectively prevent the data from being reconstructed from the feature, as we demonstrated in Section 5.3, the privacy protection and the utility can be balanced with $\lambda_2$ ranging from 2 to 4.
>
> **Thanks again for your valuable time and constructive suggestions.**

---

### Author Response · Authors · 2023-11-20
**To all reviewers: supplemented experimental results, analysis, and the revised paper uploaded**

Dear reviewers,


We sincerely thank you for your valuable time and constructive comments, which helped us improve this paper considerably. We respond to your review point by point and hope to address all your concerns. In addition, we submit a revised paper with changes highlighted in blue. We also provide more results with regard to another data modality, hyper-parameter tuning, communication efficiency, etc. We hope our efforts make the paper and the data access clearer. Please check the new materials we submit and our responses below.

---

### Meta-Review · Area_Chair_tU1u · 2023-12-11

**Metareview:**

Thank you for your submission and updating the paper. The paper introduces a new feature-sharing method in federated learning to tackle label skew and data scarcity. The authors showed improved performance and argued that their methods provide privacy protection. The reviewers identified some weaknesses of the paper. Even though the paper claims to provide privacy protection, it does not provide a rigorous evaluation or guarantee. The authors claimed that their method can be easily combined with differential privacy (DP), but also argued that there are challenges in defining DP for their setting. In general, the study of their privacy guarantees seems incomplete. We appreciate that the authors added a second dataset to their experiments. However, for a primarily empirical paper, we are hoping there will be further experiments with more diverse datasets.

**Justification For Why Not Higher Score:**

Several aspects of the paper are not satisfactory.

**Justification For Why Not Lower Score:**

N/A

---

### Decision · Program_Chairs · 2024-01-16

Reject